# Janossy Pooling: Learning Deep Permutation-invariant Functions for Variable-size Inputs

**Ryan L. Murphy**
Department of Statistics
Purdue University
murph213@purdue.edu

**Balasubramaniam Srinivasan**
Department of Computer Science
Purdue University
bsriniv@purdue.edu

**Vinayak Rao**
Department of Statistics
Purdue University
varao@purdue.edu

**Bruno Ribeiro**
Department of Computer Science
Purdue University
ribeiro@cs.purdue.edu

## Abstract

We consider a simple and overarching representation for permutation-invariant functions of sequences (or multiset functions). Our approach, which we call Janossy pooling, expresses a permutation-invariant function as the average of a permutation-sensitive function applied to all reorderings of the input sequence. This allows us to leverage the rich and mature literature on permutation-sensitive functions to construct novel and flexible permutation-invariant functions. If carried out naively, Janossy pooling can be computationally prohibitive. To allow computational tractability, we consider three kinds of approximations: canonical orderings of sequences, functions with $k$-order interactions, and stochastic optimization algorithms with random permutations. Our framework unifies a variety of existing work in the literature, and suggests possible modeling and algorithmic extensions. We explore a few in our experiments, which demonstrate improved performance over current state-of-the-art methods.

## 1 Introduction

Pooling is a fundamental operation in deep learning architectures (LeCun et al., 2015). The role of pooling is to merge a collection of related features into a single, possibly vector-valued, summary feature. A prototypical example is in convolutional neural networks (CNNs) (LeCun et al., 1995), where linear activations of features in neighborhoods of image locations are pooled together to construct more abstract features. A more modern example is in neural networks for graphs, where each layer pools together embeddings of neighbors of a vertex to form a new embedding for that vertex, see for instance, (Kipf & Welling, 2016; Atwood & Towsley, 2016; Hamilton et al., 2017; Velickovic et al., 2017; Monti et al., 2017; Xu et al., 2018; Liu et al., 2018; Liben-Nowell & Kleinberg, 2007; van den Berg et al., 2017; Duvenaud et al., 2015; Gilmer et al., 2017; Ying et al., 2018; Xu et al., 2019).

A common requirement of a pooling operator is invariance to the ordering of the input features. In CNNs for images, pooling allows invariance to translations and rotations, while for graphs, it allows invariance to graph isomorphisms. Existing pooling operators are mostly limited to pre-defined heuristics such as max-pool, min-pool, sum, or average. Another desirable characteristic of pooling layers is the ability to take variable-size inputs. This is less important in images, where neighborhoods are usually fixed *a priori*. However in applications involving graphs, the number of neighbors of different vertices can vary widely. Our goal is to design flexible and learnable pooling operators satisfying these two desiderata.

Abstractly, we will view pooling as a permutation-invariant (or symmetric) function acting on finite but arbitrary length sequences $\boldsymbol{h}$. All elements $h_i$ of the sequences are features lying in some space $\mathbb{H}$ (which itself could be a high-dimensional Euclidean space $\mathbb{R}^d$ or some subset thereof). The sequences $\boldsymbol{h}$ are themselves elements of the union of products of the $\mathbb{H}$-space: $\boldsymbol{h} \in \bigcup_{j=0}^{\infty} \mathbb{H}^j \equiv \mathbb{H}^{\cup}$. Throughout the paper, we will use $\Pi_n$ to represent the set of all permutations of the integers 1 to $n$,

where $n$ will often be clear from the context. In addition, $\boldsymbol{h}_\pi$, $\pi \in \Pi_{|\boldsymbol{h}|}$, will represent a reordering of the elements of a sequence $\boldsymbol{h}$ according to $\pi$, where $|\boldsymbol{h}|$ is the length of the sequence $\boldsymbol{h}$. We will use the double bar superscript $\overline{\overline{f}}$ to indicate that a function is permutation-invariant, returning the same value no matter the order of its arguments: $\overline{\overline{f}}(\boldsymbol{h}) = \overline{\overline{f}}(\boldsymbol{h}_\pi)$, $\forall \pi \in \Pi_{|\boldsymbol{h}|}$. We will use the arrow superscript $\vec{f}$ to indicate general functions on sequences $\boldsymbol{h}$ which may or may not be permutation-invariant[1]. Functions $f$ without any markers are 'simple' functions, acting on elements in $\mathbb{H}$, scalars or any other argument that is not a sequence of elements in $\mathbb{H}$.

Our goal in this paper is to model and learn permutation-sensitive functions $\vec{f}$ that can be used to construct flexible and learnable permutation-invariant neural networks. A recent step in this direction is work on *DeepSets* by Zaheer et al. (2017), who argued for learning permutation-invariant functions through the following composition:

$$\overline{\overline{y}}(\boldsymbol{x}; \boldsymbol{\theta}^{(\rho)}, \boldsymbol{\theta}^{(f)}, \boldsymbol{\theta}^{(h)}) = \rho\left(\overline{\overline{f}}(|\boldsymbol{h}|, \boldsymbol{h}; \boldsymbol{\theta}^{(f)}); \boldsymbol{\theta}^{(\rho)}\right), \text{where} \tag{1}$$

$$\overline{\overline{f}}(|\boldsymbol{h}|, \boldsymbol{h}; \boldsymbol{\theta}^{(f)}) = \sum_{j=1}^{|\boldsymbol{h}|} f(h_j; \boldsymbol{\theta}^{(f)}) \quad \text{and} \quad \boldsymbol{h} \equiv h(\boldsymbol{x}; \boldsymbol{\theta}^{(h)}). \tag{2}$$

Here, (a) $\boldsymbol{x} \in \mathbb{X}$ is one observation in the training data ($\mathbb{X}$ itself may contain variable-length sequences), $\boldsymbol{h} \in \mathbb{H}$ is the embedding (output) of the data given by the lower layers $h : \mathbb{X} \times \mathbb{R}^a \to \mathbb{H}^\cup$, $a > 0$ with parameters $\boldsymbol{\theta}^{(h)} \in \mathbb{R}^a$; (b) $f : \mathbb{H} \times \mathbb{R}^b \to \mathbb{F}$ is a middle-layer embedding function with parameters $\boldsymbol{\theta}^{(f)} \in \mathbb{R}^b$, $b > 0$, and $\mathbb{F}$ is the embedding space of $f$; and (c) $\rho : \mathbb{F} \times \mathbb{R}^c \to \mathbb{Y}$ is a neural network with parameters $\boldsymbol{\theta}^{(\rho)} \in \mathbb{R}^c$, $c > 0$, that maps to the final output space $\mathbb{Y}$. Typically $\mathbb{H}$ and $\mathbb{F}$ are high-dimensional real-valued spaces; $\mathbb{Y}$ is often $\mathbb{R}^d$ in $d$-dimensional regression problems or the simplex in classification problems. Effectively, the neural network $f$ learns an embedding for each element in $\mathbb{H}$, and given a sequence $\boldsymbol{h}$, its component embeddings are added together before a second neural network transformation $\rho$ is applied. Note that the function $h$ may be the identity mapping $h(\boldsymbol{x}; \cdot) = \boldsymbol{x}$ that makes $\overline{\overline{f}}$ act directly on the input data. Zaheer et al. (2017) argue that if $\rho$ is a universal function approximator, the above architecture is capable of approximating any symmetric function on $\boldsymbol{h}$-sequences, which justifies the widespread use of average (sum) pooling to make neural networks permutation-invariant in Duvenaud et al. (2015), Hamilton et al. (2017), Kipf & Welling (2016), Atwood & Towsley (2016), among other works. We note that Zaheer et al. (2017) focus on functions of sets but the work was extended to functions of multisets by Xu et al. (2019) and that Janossy pooling can be used to represent multiset functions.

In practice, there is a gap between flexibility and learnability. While the architecture of equations 1 and 2 is a universal approximator to permutation-invariant functions, it does not easily encode structural knowledge about $\overline{\overline{y}}$. Consider trying to learn the permutation-invariant function $\overline{\overline{y}}(\boldsymbol{x}) = \max_{i,j \le |\boldsymbol{x}|} |x_i - x_j|$. With higher-order interactions between the elements of $\boldsymbol{h}$, the functions $f$ of equation 2 cannot capture any useful intermediate representations towards the final output, with the burden shifted entirely to the function $\rho$. Learning $\rho$ means learning to undo mixing performed by the summation layer $\overline{\overline{f}}(|\boldsymbol{h}|, \boldsymbol{h}; \boldsymbol{\theta}^{(f)}) = \sum_{j=1}^{|\boldsymbol{h}|} f(h_j; \boldsymbol{\theta}^{(f)})$. As we show in our experiments, in many applications this is too much to ask of $\rho$.

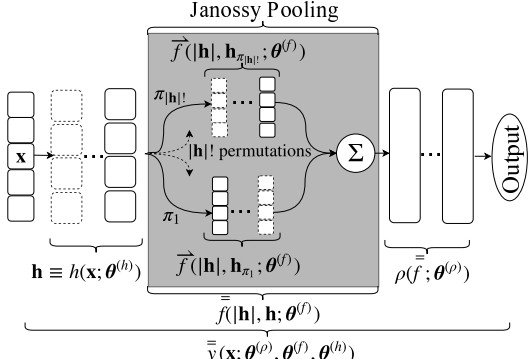

Figure 1: A neural network with a single Janossy pooling layer. The embedding $\boldsymbol{h}$ is permuted in all $|\boldsymbol{h}|!$ possible ways, and for each permutation $\boldsymbol{h}_\pi$, $\vec{f}(|\boldsymbol{h}|, \boldsymbol{h}_\pi; \boldsymbol{\theta}^{(f)})$ is computed. These are summed and passed to a second function $\rho(\cdot; \boldsymbol{\theta}^{(\rho)})$ which gives the final permutation-invariant output $\overline{\overline{y}}(\boldsymbol{x}; \boldsymbol{\theta}^{(\rho)}, \boldsymbol{\theta}^{(f)}, \boldsymbol{\theta}^{(h)})$; the gray rectangle represents Janossy pooling. We discuss how this can be made computationally tractable.

---

[1]LaTeXcode for these markers is provided in the Supplementary Material.

**Contributions.** We investigate a learnable permutation-invariant pooling layer for variable-size inputs inspired by the Janossy density framework, widely used in the theory of point processes (Daley & Vere-Jones, 2003, Chapter 7). This approach, which we call Janossy pooling, directly allows the user to model what higher-order dependencies in $\boldsymbol{h}$ are relevant in the pooling.

Figure 1 summarizes a neural network with a single Janossy pooling layer $\overline{\overline{f}}$ (detailed in Definition 2.1 below): given an input embedding $\boldsymbol{h}$, we apply a learnable (permutation-sensitive) function $\vec{f}$ to every permutation $\boldsymbol{h}_\pi$ of the input sequence $\boldsymbol{h}$. These outputs are added together, and fed to the second function $\rho$. Examples of function $\vec{f}$ include feedforward and recurrent neural networks (RNNs). We call the operation used to construct $\overline{\overline{f}}$ from $\vec{f}$ the *Janossy pooling*. Definition 2.1 gives a more detailed description. We will detail three broad strategies for making this computation tractable and discuss how existing methods can be seen as tractability strategies under the Janossy pooling framework.

Thus, we propose a framework and tractability strategies that unify and extend existing methods in the literature. We contribute the following analysis: (a) We show *DeepSets* (Zaheer et al., 2017) is a special case of Janossy pooling where the function $\vec{f}$ depends only on the first element of the sequence $\boldsymbol{h}_\pi$. In the most general form of Janossy pooling (as described above), $\vec{f}$ depends on its entire input sequence $\boldsymbol{h}_\pi$. This naturally raises the possibility of intermediate choices of $\vec{f}$ that allow practitioners to trade between flexibility and tractability. We will show that functions $\vec{f}$ that depend on their first $k$ arguments of $\boldsymbol{h}_\pi$ allow the Janossy pooling layer to capture up to $k$-ary dependencies in $\boldsymbol{h}$. (b) We show Janossy pooling can be used to learn permutation-invariant neural networks $\overline{\overline{y}}(\boldsymbol{x})$ by sampling a random permutation of $\boldsymbol{h}$ during training, and then modeling this permuted sequence using a sequence model such as a recurrent neural network (LSTMs (Hochreiter & Schmidhuber, 1997), GRUs (Cho et al., 2014)) or a vector model such as a feedforward network. We call this permutation-sampling learning algorithm $\pi$-SGD ($\pi$-Stochastic Gradient Descent). Our analysis explains why this seemingly unsound procedure is theoretically justified, which sheds light on the recent puzzling success of permutation sampling and LSTMs in relational models (Moore & Neville, 2017; Hamilton et al., 2017). We show that this property relates to randomized model ensemble techniques. (c) In Zaheer et al. (2017), the authors describe a connection between *DeepSets* and infinite de Finetti exchangeabilty. We provide a probabilistic connection between Janossy pooling and *finite* de Finetti exchangeabilty (Diaconis, 1977).

## 2 JANOSSY POOLING

We first formalize the Janossy pooling function $\overline{\overline{f}}$. Start with a function $\vec{f}$, parameterized by $\boldsymbol{\theta}^{(f)}$, which can take any variable-size sequence as input: a sequence of matrices (such as images), a sequence of vectors (such as a sequence of vector embeddings), or a variable-size sequence of features or embeddings representing the neighbors of a node in an attributed graph. In practice, we implement $\vec{f}$ with a neural network. Formalizing Figure 1 from Section 1, we use $\vec{f}$ to define $\overline{\overline{f}}$:

*Definition 2.1:* [Janossy pooling] Consider a function $\vec{f} : \mathbb{N} \times \mathbb{H}^\cup \times \mathbb{R}^b \to \mathbb{F}$ on variable-length but finite sequences $\boldsymbol{h}$, parameterized by $\boldsymbol{\theta}^{(f)} \in \mathbb{R}^b$, $b > 0$. A permutation-invariant function $\overline{\overline{f}} : \mathbb{N} \times \mathbb{H}^\cup \times \mathbb{R}^b \to \mathbb{F}$ is the Janossy function associated with $\vec{f}$ if

$$\overline{\overline{f}}(|\boldsymbol{h}|, \boldsymbol{h}; \boldsymbol{\theta}^{(f)}) = \frac{1}{|\boldsymbol{h}|!} \sum_{\pi \in \Pi_{|\boldsymbol{h}|}} \vec{f}(|\boldsymbol{h}|, \boldsymbol{h}_\pi; \boldsymbol{\theta}^{(f)}), \tag{3}$$

where $\Pi_{|\boldsymbol{h}|}$ is the set of all permutations of the integers 1 to $|\boldsymbol{h}|$, and $\boldsymbol{h}_\pi$ represents a particular reordering of the elements of sequence $\boldsymbol{h}$ according to $\pi \in \Pi_{|\boldsymbol{h}|}$. We refer the operation used to construct $\overline{\overline{f}}$ from $\vec{f}$ as Janossy pooling. $\Diamond$

Definition 2.1 provides a conceptually simple approach for constructing permutation-invariant functions from arbitrary and powerful permutation-sensitive functions such as feedforward networks, recurrent neural networks, or convolutional neural networks. If $\vec{f}$ is a vector-valued function, then so is $\overline{\overline{f}}$, and in practice, one might pass this vector output of $\overline{\overline{f}}$ through a second function $\rho$ (e.g. a

neural network parameterized by $\theta^{(\rho)}$):

$$\overline{\overline{y}}(\boldsymbol{x};\boldsymbol{\theta}^{(\rho)},\boldsymbol{\theta}^{(f)},\boldsymbol{\theta}^{(h)}) = \rho\left(\frac{1}{|\boldsymbol{h}|!}\sum_{\pi\in\Pi_{|\boldsymbol{h}|}}\vec{f}(|\boldsymbol{h}|,\boldsymbol{h}_{\pi};\boldsymbol{\theta}^{(f)});\boldsymbol{\theta}^{(\rho)}\right), \text{where} \quad \boldsymbol{h}\equiv h(\boldsymbol{x};\boldsymbol{\theta}^{(h)}). \quad (4)$$

Equation 3 can capture any permutation-invariant function $\overline{g}$ for a flexible enough family of permutation-sensitive functions $\vec{f}$ (for instance, one could always set $\vec{f} = \overline{g}$). Thus, at least theoretically, $\rho$ in equation 4 provides no additional representational power. In practice, however, $\rho$ can improve learnability by capturing common aspects across all terms in the summation. Furthermore, when we look at approximations to equation 3 or restrictions of $\vec{f}$ to more tractable families, adding $\rho$ can help recover some of the lost model capacity. Overall then, equation 4 represents one layer of Janossy pooling, forming a constituent part of a bigger neural network. Figure 1 summarizes this.

Janossy pooling, as defined in equation 3 and 4 is intractable; the computational cost of summing over all permutations (for prediction), and backpropagating gradients (for learning) is likely prohibitive for most problems of interest. Nevertheless, it provides an overarching framework to unify existing methods, and to extend them. In what follows we present strategies for mitigating this, allowing novel and effective trade-offs between learnability and computational cost.

## 2.1 Tractability through Canonical Input Orderings

A simple way to achieve permutation-invariance without the summation in equation 3 is to order the elements of $\boldsymbol{h}$ according to some canonical ordering based on its values, and then feed the reordered sequence to $\vec{f}$. More precisely, one defines a function CANONICAL : $\mathbb{H}^\cup \to \mathbb{H}^\cup$ such that CANONICAL($\boldsymbol{h}$) = CANONICAL($\boldsymbol{h}_\pi$)$\forall\pi \in \Pi_{|\boldsymbol{h}|}$ and only considers functions $\vec{f}$ based on the composition $\vec{f} = \text{CANONICAL}\circ\vec{f}'$. Note that specifying a permutation-invariant CANONICAL is not equivalent to the original problem since one may define a function of only the data and not of learnable parameters (e.g. sort). This input constraint then allows the use of complex $\vec{f}$ models, such as RNNs, that can capture arbitrary relationships in the canonical ordering of $\boldsymbol{h}$ without the need to sum over all permutations of the input.

Examples of the canonical ordering approach already exist in the literature, for example, Niepert et al. (2016) order nodes in a graph according to a user-specified ranking such as betweenness centrality (say from high to low). This approach is useful only if the canonical ordering is relevant to the task at hand. Niepert et al. (2016)acknowledges this shortcoming and Moore & Neville (2017) demonstrates that an ordering by Personalized PageRank (Page et al., 1999; Jeh & Widom, 2003) achieves a lower classification accuracy than a random ordering. As an idealized example, consider input sequences $\boldsymbol{h} = \left((h_{i,1}, h_{i,2})\right)_{i=1}^n$, with $(h_{i,1}, h_{i,2}) \in \mathbb{H} = \mathbb{R}^2$, and components $h_{i,1}$ and $h_{i,2}$ sampled independently of each other. Choosing to sort $\boldsymbol{h}$ according to $h_{\cdot,1}$ when the task at hand depends on sorting according to $h_{\cdot,2}$ can lead to poor prediction accuracy.

Rather than pre-defining a good canonical order, one can try to learn it from the data. This requires searching over the discrete space of all $|\boldsymbol{h}|!$ permutations of the input vector $\boldsymbol{h}$. In practice, this discrete optimization relies on heuristics (Vinyals et al., 2016; Rezatofighi et al., 2018). Alternatively, instead of choosing a single canonical ordering, one can choose multiple orderings, resulting in ensemble methods that average across multiple permutations. These can be viewed as more refined (possibly data-driven) approximations to equation 3.

## 2.2 Tractability through $k$-ary dependencies

Here, we provide a different spectrum of options to trade-off flexibility, complexity, and generalizability in Janossy pooling. Now, to simplify the sum over permutations in equation 3, we impose structural constraints where $\vec{f}(\boldsymbol{h})$ depends only on the first $k$ elements of its input sequence. This amounts to the assumption that only $k$-ary dependencies in $\boldsymbol{h}$ are relevant to the task at hand.

*Definition 2.2:* [$k$-ary Janossy pooling] Fix $k \in \mathbb{N}$. For any sequence $\boldsymbol{h}$, define $\downarrow_k(\boldsymbol{h})$ as its projection to a length $k$ sequence; in particular, if $|\boldsymbol{h}| \geq k$, we keep the first $k$ elements. Then, a $k$-ary

permutation-invariant Janossy function $\overline{\overline{f}}$ is given by

$$\overline{\overline{f}}(|\boldsymbol{h}|, \boldsymbol{h}; \boldsymbol{\theta}^{(f)}) = \frac{1}{|\boldsymbol{h}|!} \sum_{\pi \in \Pi_{|\boldsymbol{h}|}} \vec{f}(|\boldsymbol{h}|, \downarrow_k(\boldsymbol{h}_\pi); \boldsymbol{\theta}^{(f)}). \tag{5}$$

$$\diamondsuit$$

Note that if some of the embeddings have length $|\boldsymbol{h}| < k$, then we can zero pad to form the length-$k$ sequence $(\downarrow_k(\boldsymbol{h}_\pi), 0, \ldots, 0)$. Proposition 2.1 shows that if $|\boldsymbol{h}| > k$, equation 5 only needs to sum over $|\boldsymbol{h}|!/(|\boldsymbol{h}| - k)!$ terms, which can be tractable for small $k$.

**Proposition 2.1.** *The Janossy pooling in equation 5 requires summing over only* $\frac{|\boldsymbol{h}|!}{(|\boldsymbol{h}|-k)!}$ *terms, thus saving computation when* $k < |\boldsymbol{h}|$*. In particular, equation 5 can be written as* $\frac{(|\boldsymbol{h}|-k)!}{|\boldsymbol{h}|!} \sum_{(i_1, i_2, \ldots, i_k) \in \mathbb{I}_{|\boldsymbol{h}|}} \vec{f}(|\boldsymbol{h}|, (h_{i_1}, h_{i_2}, \ldots, h_{i_k}); \boldsymbol{\theta}^{(f)})$*, where* $\mathbb{I}_{|\boldsymbol{h}|}$ *is the set of all permutations of* $\{1, 2, \ldots, |\boldsymbol{h}|\}$ *taken* $k$ *at a time, and* $h_j$ *is the* $j$*-th element of* $\boldsymbol{h}$*.*

Note that the value of $k$ balances computational savings and the capacity to model higher-order interactions; it can be selected as a hyperparameter based on a-priori beliefs or through typical hyperparameter tuning strategies.

**Remark 2.1** (DeepSets (Zaheer et al., 2017) is a 1-ary (unary) Janossy pooling)**.** *Equation 5 represented with* $k = 1$ *and composing with* $\rho$ *as in equation 4 yields the model* $\rho\left(\frac{1}{|\boldsymbol{h}|} \sum_{j=1}^{|\boldsymbol{h}|} \vec{f}(|\boldsymbol{h}|, h_j; \boldsymbol{\theta}^{(f)}); \boldsymbol{\theta}^{(\rho)}\right)$ *and thus equations 1 and 2 for an appropriate choice of* $\vec{f}$*.*

Not surprisingly, the computational savings obtained from $k$-ary Janossy pooling come at the cost of reduced model flexibility. The next result formalizes this.

**Theorem 2.1.** *For any* $k \in \mathbb{N}$*, define* $\mathcal{F}_k$ *as the set of all permutation-invariant functions that can be represented by Janossy pooling with* $k$*-ary dependencies. Then,* $\mathcal{F}_{k-1}$ *is a* proper *subset of* $\mathcal{F}_k$ *if the space* $\mathbb{H}$ *is not trivial (i.e. if the cardinality of* $\mathbb{H}$ *is greater than 1). Thus, Janossy pooling with* $k$*-ary dependencies can express any Janossy pooling function with* $(k - 1)$*-ary dependencies, but the converse does not hold.*

The proof is given in the Supplementary Material. Theorem 2.1 has the following implication:

**Corollary 2.1.** *For* $k > 1$*, the* DeepSets *function in equation 1 (Zaheer et al., 2017) pushes the modeling of* $k$*-ary relationships to* $\rho$*.*

*Proof. DeepSets* functions can be expressed via Janossy pooling with $k = 1$. Thus, by Theorem 2.1, $\overline{\overline{f}}$ in equation 2 cannot express all functions that can be expressed by higher-order (i.e. $k > 1$) Janossy pooling operations. Consequently, if the *DeepSets* function can express any permutation-invariant function, the expressive power must have been pushed to $\rho$. □

## 2.3 Tractability through Permutation Sampling

Another approach to tractable Janossy pooling samples *random* permutations of the input $\boldsymbol{h}$ during training. Like the canonical ordering approach of Section 2.1, this offers significant computational savings, allowing more complex models for $\vec{f}$ such as LSTMs and GRUs. However, in contrast with that approach, this is considerably more flexible, avoiding the need to learn a canonical ordering or to make assumptions about the dependencies between the elements of $\boldsymbol{h}$ and the objective function. Rather, it can be viewed as *implicitly* assuming simpler structure in these functions. The approach of sampling random permutations has been previously used in relational learning tasks (Moore & Neville, 2017; Hamilton et al., 2017) as a heuristic with an LSTM as $\vec{f}$. Both these papers report that permutation sampling outperforms or closely matches other tested neural network models they tried. Therefore, this section not only proposes a tractable approximation for equation 3 but also provides a theoretical framework to understand and extend such approaches.

For the sake of simplicity, we analyze the optimization with a single sampled permutation. However, note that increasing the number of sampled permutations in the estimate of $\overline{\overline{f}}$ decreases variance, and we recover the exact algorithm when all $|\boldsymbol{h}|!$ permutations are sampled. We assume a supervised learning setting, though our analysis easily extends to unsupervised learning. We are given training data $\mathcal{D} \equiv \{(\boldsymbol{x}(1), \boldsymbol{y}(1)), \ldots, (\boldsymbol{x}(N), \boldsymbol{y}(N))\}$, where $\boldsymbol{y}(i) \in \mathbb{Y}$ is the target output and $\boldsymbol{x}(i)$ its corresponding input. Our original goal was to minimize the empirical loss

$$\overline{\overline{L}}(\mathcal{D}; \boldsymbol{\theta}^{(\rho)}, \boldsymbol{\theta}^{(f)}, \boldsymbol{\theta}^{(h)}) = \frac{1}{N} \sum_{i=1}^{N} L\left(\boldsymbol{y}(i), \rho\left(\overline{\overline{f}}(|\boldsymbol{h}^{(i)}|, \boldsymbol{h}^{(i)}; \boldsymbol{\theta}^{(f)}); \boldsymbol{\theta}^{(\rho)}\right)\right), \text{where} \tag{6}$$

$$\overline{\overline{f}}(|\boldsymbol{h}^{(i)}|, \boldsymbol{h}^{(i)}; \boldsymbol{\theta}^{(f)}) = \frac{1}{|\boldsymbol{h}^{(i)}|!} \sum_{\pi \in \Pi_{|\boldsymbol{h}^{(i)}|}} \vec{f}(|\boldsymbol{h}^{(i)}|, \boldsymbol{h}_{\pi}^{(i)}; \boldsymbol{\theta}^{(f)}) \tag{7}$$

and $\boldsymbol{h}^{(i)} = h(\boldsymbol{x}(i); \boldsymbol{\theta}^{(h)}) \in \mathbb{H}^{\cup}$ with $\boldsymbol{h}_{\pi}^{(i)} \equiv (\boldsymbol{h}^{(i)})_{\pi}$ for permutation $\pi$. Computing the gradient of equation 6 is intractable for large inputs $\boldsymbol{h}^{(i)}$, as the backpropagation computation graph branches out for every permutation in the sum. To address this computational challenge, we will turn our attention to stochastic optimization.

**Permutation sampling.** Consider replacing the Janossy sum in equation 7 with the estimate

$$\widehat{\overline{\overline{f}}}(|\boldsymbol{h}|, \boldsymbol{h}; \boldsymbol{\theta}^{(f)}) = \vec{f}(|\boldsymbol{h}|, \boldsymbol{h}_{\mathbf{s}}; \boldsymbol{\theta}^{(f)}), \tag{8}$$

where $\mathbf{s}$ is a random permutation sampled uniformly, $\mathbf{s} \sim \text{Unif}(\Pi_{|\boldsymbol{h}|})$. The estimator in equation 8 is unbiased: $E_{\mathbf{s}}[\widehat{\overline{\overline{f}}}(|\boldsymbol{h}|, \boldsymbol{h}_{\mathbf{s}}; \boldsymbol{\theta}^{(f)})] = \overline{\overline{f}}(|\boldsymbol{h}|, \boldsymbol{h}; \boldsymbol{\theta}^{(f)})$. Note however that when $\overline{\overline{f}}$ is chained with another nonlinear function $\rho$ and/or nonlinear loss $L$, the composition is no longer unbiased: $E_{\mathbf{s}}[L(\boldsymbol{y}, \rho(\vec{f}(|\boldsymbol{h}_{\mathbf{s}}|, \boldsymbol{h}_{\mathbf{s}}; \boldsymbol{\theta}^{(f)}); \boldsymbol{\theta}^{(\rho)}))] \neq L(\boldsymbol{y}, \rho(E_{\mathbf{s}}[\vec{f}(|\boldsymbol{h}_{\mathbf{s}}|, \boldsymbol{h}_{\mathbf{s}}; \boldsymbol{\theta}^{(f)})]; \boldsymbol{\theta}^{(\rho)}))$. Nevertheless, we use this estimate to propose the following stochastic approximation algorithm for gradient descent:

*Definition 2.3:* [$\pi$-SGD] Let $\mathcal{B} = \{(\boldsymbol{x}(1), \boldsymbol{y}(1)), \ldots, (\boldsymbol{x}(B), \boldsymbol{y}(B))\}$ be a mini-batch i.i.d. sampled uniformly from the training data $\mathcal{D}$. At step $t$, consider the stochastic gradient descent update

$$\boldsymbol{\theta}_t = \boldsymbol{\theta}_{t-1} - \eta_t \mathbf{Z}_t, \tag{9}$$

where $\mathbf{Z}_t = \frac{1}{B} \sum_{i=1}^{B} \nabla_{\boldsymbol{\theta}} L\left(\boldsymbol{y}(i), \rho\left(\vec{f}(|\boldsymbol{h}^{(i,t-1)}|, \boldsymbol{h}_{\mathbf{s}_i}^{(i,t-1)}; \boldsymbol{\theta}_{t-1}^{(f)}); \boldsymbol{\theta}_{t-1}^{(\rho)}\right)\right)$ is the random gradient, where $\boldsymbol{h}_{\pi}^{(i,t-1)} \equiv (h(\boldsymbol{x}(i); \boldsymbol{\theta}_{t-1}^{(h)}))_{\pi}$ for a permutation $\pi$, $\boldsymbol{\theta} \equiv (\boldsymbol{\theta}^{(\rho)}, \boldsymbol{\theta}^{(f)}, \boldsymbol{\theta}^{(h)})$, with the random permutations $\{\mathbf{s}_i\}_{i=1}^{B}$, sampled independently $\mathbf{s}_i \sim \text{Uniform}(\Pi_{|\boldsymbol{h}^{(i)}|})$; the learning rate is $\eta_t \in (0, 1)$ s.t. $\lim_{t \to \infty} \eta_t = 0$, $\sum_{t=1}^{\infty} \eta_t = \infty$, and $\sum_{t=1}^{\infty} \eta_t^2 < \infty$. $\diamond$

Effectively, this is a Robbins-Monro stochastic approximation algorithm of gradient descent (Robbins & Monro, 1951; Bottou & LeCun, 2004) and optimizes the following modified objective:

$$\overline{\overline{J}}(\mathcal{D}; \boldsymbol{\theta}^{(\rho)}, \boldsymbol{\theta}^{(f)}, \boldsymbol{\theta}^{(h)}) = \frac{1}{N} \sum_{i=1}^{N} E_{\mathbf{s}_i} \left[ L\left(\boldsymbol{y}(i), \rho\left(\vec{f}(|\boldsymbol{h}^{(i)}|, \boldsymbol{h}_{\mathbf{s}_i}^{(i)}; \boldsymbol{\theta}^{(f)}); \boldsymbol{\theta}^{(\rho)}\right)\right) \right]$$
$$= \frac{1}{N} \sum_{i=1}^{N} \frac{1}{|\boldsymbol{h}^{(i)}|!} \sum_{\pi \in \Pi_{|\boldsymbol{h}^{(i)}|}} L\left(\boldsymbol{y}(i), \rho\left(\vec{f}(|\boldsymbol{h}^{(i)}|, \boldsymbol{h}_{\pi}^{(i)}; \boldsymbol{\theta}^{(f)}); \boldsymbol{\theta}^{(\rho)}\right)\right). \tag{10}$$

Observe that the expectation over permutations is now outside the $L$ and $\rho$ functions. Like equation 6, the loss in equation 10 is also permutation-invariant, though we note that $\pi$-SGD, after a finite number of iterations, returns a $\rho(\vec{f}(\cdots, \boldsymbol{h}^{(i)}, \cdots))$ sensitive to the random input permutations of $\boldsymbol{h}^{(i)}$ presented to the algorithm. Further, unless the function $\vec{f}$ itself is permutation-invariant ($\overline{\overline{f}} = \vec{f}$), the optima of $\overline{\overline{J}}$ are different from those of the original objective function $\overline{\overline{L}}$. Instead, $\overline{\overline{J}}$ is an upper bound to $\overline{\overline{L}}$ via Jensen's inequality if $L$ is convex and $\rho$ is the identity function (equation 3); minimizing this upper bound forms a tractable surrogate to the original Janossy objective. If the function class used to model $\vec{f}$ is rich enough to include permutation-invariant functions, then the global minima of $\overline{\overline{J}}$ will include those of $\overline{\overline{L}}$. In general, minimizing the upper bound implicitly regularizes $\vec{f}$ to return functions that are insensitive to permutations of the training data. While a general $\rho$ no longer upper bounds the original objective, the implicit regularization of permutation-sensitive functions still applies to the composition $\vec{f}' \equiv \rho \circ \vec{f}$ and we show competitive results.

It is important to observe that the function $\rho$ plays a very different role in our $\pi$-SGD formulation compared to $k$-ary Janossy pooling. Previously $\rho$ was composed with an average over $\vec{f}$ to model dependencies not captured in the average– and was in some sense separate from $\vec{f}$ – whereas here it becomes absorbed directly into $\vec{f}' = \rho \circ \vec{f}$.

The next result, which we state and prove more formally in the Supplementary Material, provides some insight into the convergence properties of our algorithm. Although the conditions are difficult

to check, they are similar to those used to demonstrate the convergence of SGD, which has been empirically demonstrated to yield strong performance in practice.

**Proposition 2.2.** *[$\pi$-SGD Convergence] The optimization of $\pi$-SGD enjoys properties of almost sure convergence to the optimal $\boldsymbol{\theta}$ under similar conditions as SGD.*

**Variance reduction.**  Variance reduction of the output of a sampled permutation $\vec{f}(|\boldsymbol{h}|, \boldsymbol{h}_{\mathbf{s}}; \boldsymbol{\theta}^{(f)})$, $\mathbf{s} \sim \text{Unif}(\Pi_{|\boldsymbol{h}|})$, allows $E_{\mathbf{s}}[L(\boldsymbol{y}, \vec{f}(|\boldsymbol{h}_{\mathbf{s}}|, \boldsymbol{h}_{\mathbf{s}}; \boldsymbol{\theta}^{(f)})] \approx L(\boldsymbol{y}, E_{\mathbf{s}}[\vec{f}(|\boldsymbol{h}_{\mathbf{s}}|, \boldsymbol{h}_{\mathbf{s}}; \boldsymbol{\theta}^{(f)})])$, inducing a near-equivalence between optimizing equation 6 and equation 10.  Possible approaches include *importance sampling* (used by Chen et al. (2018b) for 1-ary Janossy), *control variates* (also used by Chen et al. (2018a) also used for 1-ary Janossy), Rao-Blackwellization (Lehmann & Casella, 2006, Section 1.7), and an output regularization, which includes a penalty for two distinct sampled permutations $\mathbf{s}$ and $\mathbf{s}'$, $\| \vec{f}(|\boldsymbol{h}|, \boldsymbol{h}_{\mathbf{s}}; \boldsymbol{\theta}^{(f)}) - \vec{f}(|\boldsymbol{h}|, \boldsymbol{h}_{\mathbf{s}'}; \boldsymbol{\theta}^{(f)})\|_2^2$, so as to reduce the variance of the sampled Janossy pooling output (used before to improve Dropout masks by Zolna et al. (2018)).

**Inference.**  The use of $\pi$-SGD to optimize the Janossy pooling layer optimizes the objective $\bar{\bar{J}}$, and thus has the following implication on how outputs should be calculated at inference time:

**Remark 2.2** (Inference). *Assume $L(\boldsymbol{y}, \hat{\boldsymbol{y}})$ is convex as a function of $\hat{\boldsymbol{y}}$ (e.g., $L$ is the $L^2$ norm, cross-entropy, or negative log-likelihood losses). At test time we estimate the output $\boldsymbol{y}(i)$ of input $\boldsymbol{x}(i)$ by computing (or estimating)*

$$\hat{\boldsymbol{y}}(\boldsymbol{x}(i)) = E_{\mathbf{s}_i}\left[\vec{f'}(|\boldsymbol{h}^{(i,\star)}|, \boldsymbol{h}_{\mathbf{s}_i}^{(i,\star)}; \boldsymbol{\theta}^{(f')\star})\right] = \frac{1}{|\boldsymbol{h}^{(i,\star)}|} \sum_{\pi \in \Pi_{|\boldsymbol{h}^{(i,\star)}|}} \vec{f'}(|\boldsymbol{h}^{(i,\star)}|, \boldsymbol{h}_{\pi}^{(i,\star)}; \boldsymbol{\theta}^{(f')\star}), \quad (11)$$

*where $\vec{f'} \equiv \rho \circ \vec{f}$, $\boldsymbol{\theta}^{(f')\star} \equiv (\boldsymbol{\theta}^{(f)\star}, \boldsymbol{\theta}^{(\rho)\star})$, $\boldsymbol{h}_{\mathbf{s}_i}^{(i,\star)} \equiv (h(\boldsymbol{x}(i); \boldsymbol{\theta}^{(h)\star}))_{\mathbf{s}_i}$ and $\boldsymbol{\theta}^{(\rho)\star}, \boldsymbol{\theta}^{(f)\star}, \boldsymbol{\theta}^{(h)\star}$ are fixed points of the $\pi$-SGD optimization. Equation 11 is a permutation-invariant function.*

**Combining $\pi$-SGD and Janossy with $k$-ary Dependencies.**  In some cases one may consider $k$-ary Janossy pooling with a moderately large value of $k$ in which case even the summation over $\frac{|\boldsymbol{h}|!}{(|\boldsymbol{h}|-k)!}$ terms (see proposition 2.1) becomes expensive.  In these cases, one may sample $\mathbf{s} \sim \text{Unif}(\Pi_{|\boldsymbol{h}|})$ and compute $\vec{\bar{\bar{f}}}_k = \vec{f}(|\boldsymbol{h}|, \downarrow_k (\boldsymbol{h}_{\mathbf{s}}); \boldsymbol{\theta}^{(f)})$ in lieu of the sum in equation 5.  Note that equation 5 defining $k$-ary Janossy pooling constitutes exact inference of a simplified model whereas $\pi$-SGD with $k$-ary dependencies constitutes approximate inference. We will return to this idea in our results section where we note that the GraphSAGE model of Hamilton et al. (2017) can be cast as a $\pi$-SGD approximation of $k$-ary Janossy pooling.

## 3 EXPERIMENTS

In what follows we empirically evaluate two tractable Janossy pooling approaches, $k$-ary dependencies (section 2.2) and sampling permutations for stochastic optimization (section 2.3), to learn permutation-invariant functions for tasks of different complexities. One baseline we compare against is DeepSets (Zaheer et al., 2017); recall that this corresponds to unary ($k = 1$) Janossy pooling (Remark 2.1). Corollary 2.1 shows that explicitly modeling higher-order dependencies during pooling simplifies the task of the upper layers ($\rho$) of the neural network, and we evaluate this experimentally by letting $k = 1, 2, 3, |\boldsymbol{h}|$ over different arithmetic tasks. We also evaluate Janossy pooling in graph tasks, where it can be used as a permutation-invariant function to aggregate the features and embeddings of the neighbors of a vertex in the graph. Note that in graph tasks, permutation-invariance is required to ensure that the neural network is invariant to permutations in the adjacency matrix (graph isomorphism). The code used to generate the results in this section are available on GitHub[2].

### 3.1 ARITHMETIC TASKS ON SEQUENCES OF INTEGERS

We first consider the task of predicting the *sum* of a sequence of integers (Zaheer et al., 2017) and extend it to predicting other permutation-invariant functions: *range*, *unique sum*, *unique count*, and *variance*. In the *sum* task we predict the sum of a sequence of 5 integers drawn uniformly at random with replacement from $\{0, 1, \ldots, 99\}$; the *range* task also receives a sequence 5 integers distributed the same way and tries to predict the range (the difference between the maximum and

---

[2] https://github.com/PurdueMINDS/JanossyPooling

minimum values); the *unique sum* task receives a sequence of 10 integers, sampled uniformly with replacement from $\{0, 1, \ldots, 9\}$, and predicts the sum of all unique elements; the *unique count* task also receives a sequence of repeating elements from $\{0, 1, \ldots, 9\}$, distributed in the same was as with the *unique sum* task, and predicts the number of unique elements; the *variance* task receives a sequence of 10 integers drawn uniformly with replacement from $\{0, 1, \ldots, 99\}$ and tries to predict the variance $\frac{1}{|\boldsymbol{x}|} \sum_i (x_i - \bar{\boldsymbol{x}})^2 = \frac{1}{2|\boldsymbol{x}|^2} \sum_{i,j} (x_i - x_j)^2$, where $\bar{\boldsymbol{x}}$ denotes the mean of $\boldsymbol{x}$. Unlike Zaheer et al. (2017), we choose to work with the digits themselves, to allow a more direct assessment of the different Janossy pooling approximations. Note that the summation task of Zaheer et al. (2017) is naturally a unary task that lends itself to the approach of embedding individual digits then adding them together while the other tasks require exploiting high-order relationships within the sequence. Following Zaheer et al. (2017), we report accuracy (0-1 loss) for all tasks with an integer target; we report root mean squared error (RMSE) for the variance task.

Here we explore two Janossy pooling tractable approximations:
(a) (*k*-ary dependencies) *Janossy* ($k = 1$) *(DeepSets)*, and *Janossy* $k = 2, 3$ where $\vec{f}$ is a feedforward network with a single hidden layer comprised of 30 neurons. As detailed in the Supplementary Material, the models are constructed to have the same number of parameters regardless of $k$ by modifying the embedding (output) dimension of $h$. In the Supplementary Material, we also show results for experiments that relax this constraint.
(b) ($\pi$-SGD) Full $k = |\boldsymbol{h}|$ Janossy pooling where $\vec{f}$ is an LSTM or a GRU that returns the short-term hidden state of the last temporal unit (the $\boldsymbol{h}_t$ of Cho et al. with $t = |\boldsymbol{h}|$). The LSTM has 50 hidden units and the GRU 80, trained with the $\pi$-SGD stochastic optimization. The number of hidden units was chosen to be consistent with Zaheer et al. (2017). At test time, we experiment with approximating (estimating) equation 11 using 1 and 20 sampled permutations.

We also explore two functions for $\rho$ of equation 4 (upper-layer): (i) [Linear] a single dense layer with identity activation as in the experiments of Zaheer et al. (2017), and (ii) [MLP (100)] a feedforward network with one hidden layer using tanh activations and 100 units. Choosing a simple and complex form for $\rho$ allows insight into the extent to which $\rho$ supplements the capacity of the model by capturing relationships not exploited during pooling, and serves as an evaluation of the strategy of optimizing $\bar{\bar{J}}$ as a tractable approximation of $\bar{\bar{L}}$.

Much of our implementation, architectural, and experimental design are based on the DeepSets code[3] of Zaheer et al. (2017), see the Supplementary Material for details. We tuned the Adam learning rate for each model and report the results using the rate yielding top performance on the validation set. Table 1 shows the accuracy (average 0-1 loss) of all tasks except variance, for which we report RMSE in the last column. Performance was similar between the LSTM and GRU models, with the GRU performing slightly better, thus we moved the LSTM results to Table 3 in the Supplementary Material for the sake of clarity. We trained each model with 15 random initializations of the weights to quantify variability. Table 4 in the Supplementary Material shows the same results measured by mean absolute error. The data consists of 100,000 training examples and 10,000 test examples.

The results in Table 1 and Table 3 show that: (1) models trained with $\pi$-SGD using LSTMs and GRUs as $\vec{f}$ typically achieve top performance or are comparable to the top performer (within confidence intervals) on all tasks, for any choice of $\rho$. We also observe for LSTMs and GRUs that adding complexity to $\rho$ can yield small but meaningful performance gains or maintain similar performance, lending credence to the approach of optimizing $\bar{\bar{J}}$ as a tractable approximation to $\bar{\bar{L}}$. (2) Specifically, in the *variance* task, GRUs and LSTMs with $\pi$-SGD provide significant accuracy gains over $k \in \{1, 2, 3\}$, showing that modeling full-dependencies can be advantageous even if model training with $\pi$-SGD is approximate. (3) For a more complex $\rho$ (MLP as opposed to Linear), lower-complexity Janossy pooling achieves consistently better results: $k \in \{2, 3\}$ gives good results when $\rho$ is linear but poorer results when $\rho$ is an MLP (as these models are more expressive, the only feasible explanation is an optimization issue since we also observed poorer performance on the *training data*). We also note that when $\rho$ is an MLP, it takes significantly more epochs for $k \in \{2, 3\}$ to find the best model (2000 epochs) while $k = 1$ finds good models much quicker (1000 epochs). The results we report come from training with 1000 epochs on all models with a linear $\rho$ and 2000 epochs for all models where $\rho$ is an MLP. (4) We observe that for $k = 1$ (DeepSets), a more complex

---

[3]https://github.com/manzilzaheer/DeepSets

Table 1: Accuracy (and RMSE for the *variance* task) of various Janossy pooling approximations under distinct tasks. The *method* column refers to the method used to deal with the sum over all permutations. *Infr sample* refers to the number of permutations sampled at test time to estimate equation 11 for methods learned with $\pi$-SGD. $k = 1$ corresponds to DeepSets. `tanh` activations are used with the MLP. Standard deviations computed over 15 runs are shown in parentheses.

| $\vec{f}$ | method | infr sample | k | $\rho$ | sum | range | unique sum | uniq. count | variance |
|---|---|---|---|---|---|---|---|---|---|
| MLP (30) | exact | – | 1 | Linear | 1.00(0.00) | 0.04(0.00) | 0.07(0.00) | 0.36(0.01) | 119.05(1.29) |
| MLP (30) | exact | – | 2 | Linear | 0.99(0.00) | 0.09(0.00) | 0.17(0.00) | 0.74(0.03) | 4.37(0.50) |
| MLP (30) | exact | – | 3 | Linear | 0.99(0.00) | 0.21(0.00) | 0.44(0.02) | 0.89(0.04) | 8.99(0.99) |
| MLP (30) | exact | – | 1 | MLP (100) | 1.00(0.00) | 0.97(0.01) | 1.00(0.00) | 1.00(0.00) | 1.95(0.24) |
| MLP (30) | exact | – | 2 | MLP (100) | 1.00(0.00) | 0.97(0.01) | 1.00(0.00) | 1.00(0.00) | 3.49(0.48) |
| MLP (30) | exact | – | 3 | MLP (100) | 0.93(0.02) | 0.93(0.02) | 1.00(0.00) | 1.00(0.00) | 6.90(0.47) |
| GRU(80) | $\pi$-SGD | 1 | $|\boldsymbol{h}|$ | Linear | 0.99(0.01) | 0.98(0.00) | 1.00(0.00) | 1.00(0.00) | 1.43(0.23) |
| GRU(80) | $\pi$-SGD | 20 | $|\boldsymbol{h}|$ | Linear | 0.99(0.00) | 0.99(0.00) | 1.00(0.00) | 1.00(0.00) | 1.20(0.23) |
| GRU(80) | $\pi$-SGD | 1 | $|\boldsymbol{h}|$ | MLP (100) | 0.99(0.00) | 1.00(0.00) | 1.00(0.00) | 1.00(0.00) | 0.42(0.62) |
| GRU(80) | $\pi$-SGD | 20 | $|\boldsymbol{h}|$ | MLP (100) | 0.99(0.00) | 1.00(0.00) | 1.00(0.00) | 1.00(0.00) | 0.40(0.37) |

$\rho$ (MLP) is required as the pooling pushes the complexity of modeling high-order interactions over the input to $\rho$. The converse is also true, if $\rho$ is simple (Linear) then a Janossy pooling that models high-order interactions $k \in \{2, 3, |\boldsymbol{h}|\}$ gives higher accuracy, as shown in the *range*, *unique sum*, *unique count*, and *variance* tasks.

## 3.2 JANOSSY POOLING AS AN AGGREGATOR FUNCTION FOR VERTEX CLASSIFICATION

Here we consider Janossy pooling in the context of graph neural networks to learn vertex representations enabling vertex classification. The GraphSAGE algorithm (Hamilton et al., 2017) consists of sampling vertex attributes from the neighbor multiset of each vertex $v$ before performing an aggregation operation which generates an embedding of $v$; the authors consider permutation-invariant operations such as mean and max as well as the permutation-sensitive operation of feeding a randomly permuted neighborhood sequence to an LSTM. The sample and aggregate procedure is repeated twice to generate an embedding. Each step can be considered as Janossy pooling with $\pi$-SGD and $k$-ary subsequences, where $k_l$, $l \in \{1, 2\}$ is the number of vertices sampled from each neighborhood and $\vec{f}$ is for instance a mean, max, or LSTM. However, at test time, GraphSAGE only samples one permutation **s** of each neighborhood to estimate equation 11.

In our experiments, we also consider computing the mean of the entire neighborhood. Here we say $k = 1$ to reinforce the connection to unary Janossy pooling whereas with the LSTM model, $k$ refers to the number of samples of the neighborhood.

In this section we investigate two conditions: (a) the impact of increasing $k$ in the $k$-ary dependencies; and (b) the benefits of increasing the number of sampled permutations at inference time. To implement the model and design our experiments, we modified the reference PyTorch code provided by the authors[4]. We consider the three graph datasets considered in Hamilton et al. (2017): Cora and Pubmed (Sen et al., 2008) and the larger Protein-Protein Interaction (PPI) (Zitnik & Leskovec, 2017). The first two are citation networks where vertices represent papers, edges represent citations, and vertex features are bag-of-words representations of the document text. The task is to classify the paper topic. The PPI dataset is a collection of several graphs each representing human tissue; vertices represent proteins, edges represent protein interaction, features include genetic and immunological features, and we try to classify protein roles (there are 121 targets). More details of these experiments are shown in Table 9 in the Supplementary Material.

(a) Table 2 shows the impact (on accuracy) of increasing the number of $k$-ary dependencies. We use $k_1, k_2 \in \{3, 5, 10, 25\}$ for the two pooling layers of our graph neural network (GNN). The function $\vec{f}$ is an LSTM (except for when we try mean-pooling). Note that for the LSTM, the number of parameters of the model is independent of $k$. At inference time, we sample 20 random permutations of each sequence and average the predicted probabilities before making a final prediction of the class label. The results in Table 2 show that the choice of $k_1, k_2 \in \{3, 5, 10, 25\}$ makes little difference on Cora and Pubmed due to the small neighborhood sizes: $k_1, k_2 \geq 5$ often amounts to

---

[4] `https://github.com/williamleif/graphsage-simple/`, see Appendix for details.

Table 2: Accuracy (Micro-F1 score) using Janossy pooling with $k$-ary dependencies and $\pi$-SGD in a graph neural network – GraphSAGE – with 20 permutations sampled at test time. Standard deviations over 30 runs for Cora/Pubmed and 4 runs for PPI are shown in parentheses.

| $\vec{f}$ | method | $k_1$ | $k_2$ | **CORA** | **PUBMED** | **PPI** |
|---|---|---|---|---|---|---|
| LSTM | $\pi$-SGD | 3 | 3 | 0.860 (.009) | 0.889 (0.01) | 0.538 (.005) |
| LSTM | $\pi$-SGD | 5 | 5 | –[a] | –[a] | 0.579 (.015) |
| LSTM | $\pi$-SGD | 10 | 25 | –[a] | –[a] | 0.650 (.013) |
| LSTM | $\pi$-SGD | 25 | 10 | –[a] | –[a] | 0.689 (.062) |
| LSTM | $\pi$-SGD | 25 | 25 | –[a] | –[a] | 0.702 (.044) |
| LSTM | $\pi$-SGD | $|\boldsymbol{h}|$ | $|\boldsymbol{h}|$ | –[a] | –[a] | 0.757 (.040) [b] |
| Identity (mean-pool) | exact | 1 | 1 | 0.860 (.008) | 0.881 (.011) | 0.767 (.013) |

[a] Entries denoted by – all differ by less than 0.01. Typical neighborhoods in Cora and Pubmed are small, so that sampling $\geq 5$ neighbors is often equivalent to using the entire neighborhood.
[b] Some neighbor sequences in PPI are prohibitively large, so we take $k_1 = k_2 = 100$.

sampling the entire neighborhood. In PPI, whose average degree is 28.8, increasing $k$ yields consistent improvement. The strong performance of mean-pooling points to both a relatively easy task[5] and the benefits of utilizing the entire neighborhood of each vertex. (b) We now investigate whether increasing the number of sampled permutations used to estimate equation 11 at test (inference) time impacts accuracy. Figure 2 in the Supplementary Material shows that increasing the number of sampled permutations from one to three leads to an increase in accuracy in the PPI task (Cora and Pubmed degrees are too small for this test) but diminishing returns set in by the seventh sample. Using paired tests – t and Wilcoxon signed rank – we see that test inference with seven sampled permutations versus one permutation is significant with $p < 10^{-3}$ over 12 replicates. Sampling permutations at inference time is thus a cheap method for achieving modest but potentially important gains at inference time.

## 4 RELATED WORK

Under the Janossy pooling framework presented in this work, existing literature falls under one of three approaches to approximating to the intractable Janossy-pooling layer: *Canonical orderings*, *$k$-ary dependencies*, and *permutation sampling*. We also discuss the broader context of invariant models and probabilistic interpretations.

**Canonical Ordering Approaches.** In section 2.1, we saw how permutation invariance can be achieved by mapping permutations to a canonical ordering. Rather than trying to define a good canonical ordering, one can try to learn it from the data, however searching among all $|\boldsymbol{h}|!$ permutations for one that correlates with the task of interest is a difficult discrete optimization problem. Recently, Rezatofighi et al. (2018) proposed a method that computes the posterior distribution of all permutations, conditioned on the model and the data. This posterior-sampling approach is intractable for large inputs, unfortunately. We note in passing that Rezatofighi et al. (2018) is interested in permutation-invariant outputs, and that Janossy pooling is also trivially applicable to these tasks. Vinyals et al. (2016) proposes a heuristic using ancestral sampling while learning the model.

**$k$-ary Janossy Pooling Approaches.** In section 2.2 we described $k$-ary Janossy pooling, which considers $k$-order relationships in the input vector $\boldsymbol{h}$ to simplify optimization. DeepSets (Zaheer et al., 2017) can be characterized as unary Janossy pooling (i.e., $k$-ary for $k = 1$). . Qi et al. (2017) and Ravanbakhsh et al. (2017a) propose similar unary Janossy pooling models. Cotter et al. (2018) proposes to add inductive biases to the *DeepSets* model in the form of monotonicity constraints with respect to the vector valued elements of the input sequence by modeling $f$ and $\rho$ with Deep Lattice Networks (You et al., 2017); one can extend Cotter et al. (2018) by using higher-order ($k > 1$) pooling.

Exploiting dependencies within a sequence to learn a permutation-invariant function has been discussed elsewhere. For instance Santoro et al. (2017) exploits pairwise relationships to perform relational reasoning about pairs of objects in an image and Battaglia et al. (2018) contemplates modeling the center of mass of a solar system by including the pairwise interactions among planets. However, Janossy pooling provides a general framework for capturing dependencies within a permutation-invariant pooling layer.

---

[5]The topic of a paper can be adequately predicted by computing the average bag-of-words representations of papers in the neighborhood without reasoning about relationships between neighboring papers.

**Permutation Sampling Approaches.** In section 2.3 we have seen a that permutation sampling can be used as a stochastic gradient procedure ($\pi$-SGD) to learn a model with a Janossy pooling layer. The learned model provides only an approximate solution to original permutation-invariant function. Permutation sampling has been used as a heuristic (without a theoretical justification) in both Moore & Neville (2017) and Hamilton et al. (2017), which found that randomly permuting sequences and feeding them forward to an LSTM is effective in relational learning tasks that require permutation-invariant pooling layers.

**Probabilistic Interpretation and Other Invariances** Our work has a strong connection with finite exchangeability. Some researchers may be more familiar with the concept of *infinite exchangeability* through de Finetti's theorem (De Finetti, 1937; Diaconis, 1977), which imposes strong structural requirements: the probability of any subsequence must equal the marginalized probability of the original sequence (projectivity). Korshunova et al. (2018) noted the importance of this property for generative models and propose a model that learns a distribution without variational approximations. Finite exchangeability drops this projectivity requirement (Diaconis, 1977), which in general, cannot be simplified beyond first sampling the number of observations $m$, and then sampling their locations from some exchangeable but non-i.i.d. distribution $p_{\text{exch}}^m(x_1, \ldots, x_m)$ (Daley & Vere-Jones, 2003). Equivalently, de Finetti's theorem for infinitely exchangeable sequences implies that the joint distribution can represented as a mixture distribution over conditionally independent random variables (given $\theta$) (De Finetti, 1937; Orbanz & Roy, 2015) whereas the probability distribution of a finitely exchangeability sequence is a mixture over *dependent* random variables as shown by Diaconis (1977).

In comparison, the restrictive assumption of letting $k = 1$ in $k$-ary Janossy Pooling yields the form of a log-likelihood of conditionally iid random variables (consider $\vec{f}$ a log pdf), the strong requirement of de Finetti's theorem for infinitely exchangeable sequences. Conversely, higher-order Janossy pooling was designed to exploit dependencies among the random variables such as those that arise under finitely exchangeable distributions. Indeed, finite exchangeability also arises from the theory of spatial point processes; our framework of Janossy pooling is inspired by *Janossy densities* (Daley & Vere-Jones, 2003), which model the finite exchangeable distributions as mixtures of non-exchangeable distributions applied to permutations. This literature also studies simplified exchangeable point processes such as finite Gibbs models (Vo et al., 2018; Moller & Waagepetersen, 2003) that restrict the structure of $p_{\text{exch}}$ to fixed-order dependencies, and are related to $k$-ary Janossy.

More broadly, there are other connections between permutation-invariant deterministic functions and exchangeability in probability distributions, as recently discussed by Bloem-Reddy & Teh (2019). There, the authors also contemplate more general invariances through the language of group actions. An example is *permutation equivariance*: one form of permutation equivariance asserts that $f(X_\pi) = f(X)_\pi \forall \pi \in \Pi_{|X|}$ where $f(X)$ is a sequence of length greater than 1. Ravanbakhsh et al. (2017b) provides a weight-sharing scheme for maintaining general neural network equivariances characterized as automorphisms of a colored multi-edged bipartite graph. Hartford et al. (2018) proposes a matrix completion model invariant to (possibly separate) permutations of the rows or columns. Other invariances are studied through a probabilistic perspective in Orbanz & Roy (2015).

## 5 CONCLUSIONS

Our approach of permutation-invariance through Janossy pooling unifies a number of existing approaches, and opens up avenues to develop both new methodological extensions, as well as better theory. Our paper focused on two main approaches: $k$-ary interactions and random permutations. The former involves exact Janossy pooling for a restricted class of functions $\vec{f}$. Adding an additional neural network $\rho$ can recover lost model capacity and capture additional higher-order interactions, but hurts tractability and identifiability. Placing restrictions on $\rho$ (convexity, Lipschitz continuity etc.) can allow a more refined control of this trade-off, allowing theoretical and empirical work to shed light on the compromises involved. The second was a random permutation approach which conversely involves no clear trade-offs between model capacity and computation when $\rho$ is made more complex, instead it modifies the relationship between the tractable approximate loss $\overline{\overline{J}}$ and the original Janossy loss $\overline{L}$. While there is a difference between $\overline{\overline{J}}$ and $\overline{L}$, we saw the strongest empirical performance coming from this approach in our experiments (shown in the last row of Table 1); future work is required to identify which problems $\pi$-SGD is best suited for and when its conver-

gence criteria are satisfied. Further, a better understanding how the loss-functions $\overline{\overline{L}}$ and $\overline{\overline{J}}$ relate to each other can shed light on the slightly black-box nature of this procedure. It is also important to understand the relationship between the random permutation optimization to canonical ordering and how one might be used to improve the other. Finally, it is important to apply our methodology to a wider range of applications. Two immediate domains are more challenging tasks involving graphs and tasks involving non-Poisson point processes.

ACKNOWLEDGMENTS

This work was sponsored in part by the ARO, under the U.S. Army Research Laboratory contract number W911NF-09-2-0053, by the Purdue Integrative Data Science Initiative and Purdue Research foundation, the DOD through SERC under Contract No. HQ0034-13-D-0004 RT #206, the National Science Foundation under contract numbers IIS-1816499 and DMS-1812197, and the NVIDIA GPU grant program for hardware donation.

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

## A  PROOFS OF RESULTS

We restate and prove Proposition 2.1.

**Proposition 2.1.** *The Janossy pooling in equation 5 requires summing over only $\frac{|\boldsymbol{h}|!}{(|\boldsymbol{h}|-k)!}$ terms, thus saving computation when $k < |\boldsymbol{h}|$. In particular, equation 5 can be written as $\frac{(|\boldsymbol{h}|-k)!}{|\boldsymbol{h}|!} \sum_{(i_1,i_2,\ldots,i_k)\in\mathbb{I}_{|\boldsymbol{h}|}} \vec{f}\left(|\boldsymbol{h}|, (h_{i_1}, h_{i_2}, \ldots, h_{i_k}); \boldsymbol{\theta}^{(f)}\right)$, where $\mathbb{I}_{|\boldsymbol{h}|}$ is the set of all permutations of $\{1, 2, \ldots, |\boldsymbol{h}|\}$ taken $k$ at a time, and $h_j$ is the $j$-th element of $\boldsymbol{h}$.*

*Proof.* Define two permutations $\pi, \pi' \in \Pi_{|\boldsymbol{h}|}$ that agree on the first $k$ elements as *$k$-equivalent*. Such permutations satisfy $\vec{f}(|\boldsymbol{h}|, \downarrow_k(\boldsymbol{h}_\pi); \boldsymbol{\theta}^{(f)}) = \vec{f}(|\boldsymbol{h}|, \downarrow_k(\boldsymbol{h}_{\pi'}); \boldsymbol{\theta}^{(f)})$. These two permutations belong to the same equivalence class, containing a total of $(|\boldsymbol{h}| - k)!$ permutations (obtained by permuting the last $(|\boldsymbol{h}| - k)$ elements). Overall, we then have a total of $|\boldsymbol{h}|!/(|\boldsymbol{h}| - k)!$ equivalence classes. Write the set of equivalence classes as $\Pi^k_{|\boldsymbol{h}|}$, and represent each by one of its elements. Then,

$$\overline{\overline{f}}(|\boldsymbol{h}|, \boldsymbol{h}; \boldsymbol{\theta}^{(f)}) = \frac{1}{|\boldsymbol{h}|!} \sum_{\pi\in\Pi_{|\boldsymbol{h}|}} \vec{f}(|\boldsymbol{h}|, \downarrow_k(\boldsymbol{h}_\pi); \boldsymbol{\theta}^{(f)}) = \frac{(|\boldsymbol{h}|-k)!}{|\boldsymbol{h}|!} \sum_{\pi\in\Pi^k_{|\boldsymbol{h}|}} \vec{f}\left(|\boldsymbol{h}|, \downarrow_k(\boldsymbol{h}_\pi); \boldsymbol{\theta}^{(f)}\right)$$

is now a summation over only $|\boldsymbol{h}|!/(|\boldsymbol{h}| - k)!$ terms. We can conclude that

$$\overline{\overline{f}}(|\boldsymbol{h}|, \boldsymbol{h}; \boldsymbol{\theta}^{(f)}) = \frac{(|\boldsymbol{h}| - k)!}{|\boldsymbol{h}|!} \sum_{(i_1, i_2, \ldots, i_k) \in \mathbb{I}_{|\boldsymbol{h}|}} \vec{f}(|\boldsymbol{h}|, (h_{i_1}, h_{i_2}, \ldots, h_{i_k}); \boldsymbol{\theta}^{(f)}).$$

$\square$

Next, we restate and prove the remaining portion of Theorem 2.1.

**Theorem 2.1.** *For any $k \in \mathbb{N}$, define $\mathcal{F}_k$ as the set of all permutation-invariant functions that can be represented by Janossy pooling with $k$-ary dependencies. Then, $\mathcal{F}_{k-1}$ is a* proper *subset of $\mathcal{F}_k$ if the space $\mathbb{H}$ is not trivial (i.e. if the cardinality of $\mathbb{H}$ is greater than 1). Thus, Janossy pooling with $k$-ary dependencies can express any Janossy pooling function with $(k-1)$-ary dependencies, but the converse does not hold.*

*Proof.*

$(\mathcal{F}_{k-1} \subset \mathcal{F}_k)$: Consider any element $\overline{\overline{f}}_{k-1} \in \mathcal{F}_{k-1}$, and write $\vec{f}(|\boldsymbol{h}|, \cdot ; \boldsymbol{\theta}^{(f)})$ for its associated Janossy function. For any sequence $\boldsymbol{h}$, $\vec{f}(|\boldsymbol{h}|, \downarrow_{k-1}(\boldsymbol{h}); \boldsymbol{\theta}^{(f)}) = \vec{f}(|\boldsymbol{h}|, \downarrow_{k-1}(\downarrow_k(\boldsymbol{h})); \boldsymbol{\theta}^{(f)}) := \vec{f}_+(|\boldsymbol{h}|, \downarrow_k(\boldsymbol{h}); \boldsymbol{\theta}^{(f)})$, where the function $\vec{f}_+$ looks at its first $k$ elements. Thus,

$$\overline{\overline{f}}_{k-1}(|\boldsymbol{h}|, \boldsymbol{h}; \boldsymbol{\theta}^{(f)}) = \frac{1}{|\boldsymbol{h}|!} \sum_{\pi \in \Pi_{|\boldsymbol{h}|}} \vec{f}(|\boldsymbol{h}|, \downarrow_{k-1}(\boldsymbol{h}_\pi); \boldsymbol{\theta}^{(f)}) = \frac{1}{|\boldsymbol{h}|!} \sum_{\pi \in \Pi_{|\boldsymbol{h}|}} \vec{f}_+(|\boldsymbol{h}|, \downarrow_k(\boldsymbol{h}_\pi); \boldsymbol{\theta}^{(f)})$$

$$= \overline{\overline{f}}_k(|\boldsymbol{h}|, \boldsymbol{h}; \boldsymbol{\theta}^{(f)}), \tag{12}$$

where $\overline{\overline{f}}_k$ is the Janossy function associated with $\vec{f}_+$ and thus belongs to $\mathcal{F}_k$.

$(\mathcal{F}_k \not\subset \mathcal{F}_{k-1})$: the case where $k = 1$ is trivial, so assume $k > 1$. We will demonstrate the existence of $\overline{\overline{f}}_k \in \mathcal{F}_k$ such that $\overline{\overline{f}}_{k-1} \neq \overline{\overline{f}}_k$ for all $\overline{\overline{f}}_{k-1} \in \mathcal{F}_{k-1}$. Let $\overline{\overline{f}}_k$ and $\overline{\overline{f}}_{k-1}$ be associated with $\vec{f}_k$ and $\vec{f}_{k-1}$, respectively.

It suffices to consider $|\boldsymbol{h}| = k$. Let $\vec{f}_k(|\boldsymbol{h}|, \boldsymbol{h}_\pi; \boldsymbol{\theta}_k^{(f)}) = \prod_{l=1}^{|\boldsymbol{h}|} h_{\pi(l)}$ whence $\overline{\overline{f}}_k(|\boldsymbol{h}|, \boldsymbol{h}; \boldsymbol{\theta}_k^{(f)}) = \prod_{l=1}^{|\boldsymbol{h}|} h_l$. Thus, for any $\overline{\overline{f}}_{k-1}$ and any $\boldsymbol{\theta}_{k-1}^{(f)}$,

$$\frac{\overline{\overline{f}}_{k-1}(|\boldsymbol{h}|, \boldsymbol{h}; \boldsymbol{\theta}_{k-1}^{(f)})}{\overline{\overline{f}}_k(|\boldsymbol{h}|, \boldsymbol{h}; \boldsymbol{\theta}_k^{(f)})} = \frac{1}{|\boldsymbol{h}|!} \sum_{\pi \in \Pi_{|\boldsymbol{h}|}} \frac{\vec{f}_{k-1}(|\boldsymbol{h}|, \downarrow_{k-1}(\boldsymbol{h}_\pi); \boldsymbol{\theta}_{k-1}^{(f)})}{\prod_{l=1}^{|\boldsymbol{h}|} h_l}$$

$$= \frac{1}{|\boldsymbol{h}|!} \sum_{j=1}^{|\boldsymbol{h}|} \sum_{\tilde{\pi} \in \Pi_{\{1, \ldots, |\boldsymbol{h}|\} \setminus j}} \frac{\vec{f}_{k-1}(|\boldsymbol{h}|, (\boldsymbol{h}_{-j})_{\tilde{\pi}}; \boldsymbol{\theta}_{k-1}^{(f)})}{\prod_{l=1}^{|\boldsymbol{h}|} h_l}$$

where $\Pi_{\{1, \ldots, |\boldsymbol{h}|\} \setminus j}$ denotes the set of permutation functions defined on $\{1, 2, \ldots, j-1, j+1, \ldots, |\boldsymbol{h}|\}$ and $(\boldsymbol{h}_{-j})_{\tilde{\pi}}$ is a permutation of the sequence $(h_1, \ldots, h_{j-1}, h_{j+1}, \ldots, h_{|\boldsymbol{h}|})$. This can be written as

$$\frac{1}{|\boldsymbol{h}|!} \sum_{j=1}^{|\boldsymbol{h}|} \frac{1}{h_j} \underbrace{\left( \sum_{\tilde{\pi} \in \Pi_{\{1, \ldots, |\boldsymbol{h}|\} \setminus j}} \frac{\vec{f}_{k-1}(|\boldsymbol{h}|, (\boldsymbol{h}_{-j})_{\tilde{\pi}}; \boldsymbol{\theta}_{k-1}^{(f)})}{\prod_{l \neq j} h_l} \right)}_{\text{denote by } a_{j, |\boldsymbol{h}|}},$$

therefore

$$\frac{\overline{\overline{f}}_{k-1}(|\boldsymbol{h}|, \boldsymbol{h}_\pi; \boldsymbol{\theta}_{k-1}^{(f)})}{\overline{\overline{f}}_k(|\boldsymbol{h}|, \boldsymbol{h}_\pi; \boldsymbol{\theta}_k^{(f)})} = \frac{1}{|\boldsymbol{h}|!} \sum_{j=1}^{|\boldsymbol{h}|} \frac{1}{h_j} a_{j, |\boldsymbol{h}|}. \tag{13}$$

Now, $\overline{\overline{f}}_{k-1} = \overline{\overline{f}}_k$ if and only if their quotient in equation 13 is unity for all $\boldsymbol{h}$. But this is clearly not possible in general unless $\mathbb{H}$ is a singleton, which we have precluded in our assumptions. $\square$

Proposition 2.2 is repeated below and is followed by a more rigorous restatement.

**Proposition 2.2.** *[$\pi$-SGD Convergence] The optimization of $\pi$-SGD enjoys properties of almost sure convergence to the optimal $\boldsymbol{\theta}$ under similar conditions as SGD.*

The following statement is similar to that in Yuille (2004), which also provides intuition behind the theoretical assumptions, which are indeed quite general. See also (Younes, 1999). This is a familiar application of stochastic approximation algorithms already used in training neural networks.

**Proposition A.1** ($\pi$-SGD Convergence). *Consider the $\pi$-SGD algorithm in Definition 2.3. If*

(a) *there exists a constant $M > 0$ such that for all $\boldsymbol{\theta}$, $-\boldsymbol{G}_t^T \boldsymbol{\theta} \leq M\|\boldsymbol{\theta} - \boldsymbol{\theta}^\star\|_2^2$, where $\boldsymbol{G}_t$ is the true gradient for the full batch over all permutations, $\boldsymbol{G}_t = \nabla_{\boldsymbol{\theta}} \overline{\overline{J}}(\mathcal{D}; \boldsymbol{\theta}_t^{(\rho)}, \boldsymbol{\theta}_t^{(f)}, \boldsymbol{\theta}_t^{(h)})$, where $\boldsymbol{\theta} \equiv (\boldsymbol{\theta}^{(\rho)}, \boldsymbol{\theta}^{(f)}, \boldsymbol{\theta}^{(h)})$, and $\boldsymbol{\theta}^\star$ is the optimum.*

(b) *there exists a constant $\delta > 0$ such that for all $\boldsymbol{\theta}$, $E_t[\|\mathbf{Z}_t\|_2^2] \leq \delta^2(1 + \|\boldsymbol{\theta}_t - \boldsymbol{\theta}_t^\star\|_2^2)$, where the expectation is taken with respect to all the data prior to step $t$.*

*Then, the algorithm in equation 9 converges to $\boldsymbol{\theta}^\star$ with probability one.*

*Proof.* First, we can show that $E_t[\mathbf{Z}_t] = \boldsymbol{G}_t$ by equation 10, the linearity of the derivative operator, and the fact that the permutations are independently sampled for each training example in the mini-batch and are assumed independent of $\boldsymbol{\theta}$. That equation 9 converges to $\boldsymbol{\theta}^\star$ is a consequence of our conditions and the supermartingale convergence theorem (Grimmett & Stirzaker, 2001, pp. 481). The following argument follows Yuille (2004). Let $A_t = \|\boldsymbol{\theta}_t - \boldsymbol{\theta}^\star\|_2^2$, $B_t = \delta^2\eta_t^2$, and $C_t = -\|\boldsymbol{\theta}_t - \boldsymbol{\theta}^\star\|_2^2(\delta^2\eta_t^2 - 2M\eta_t)$. Note that $C_t$ is positive for a sufficiently large $t$, and $\sum_{t=1}^\infty B_t \leq \infty$ by our definition of $\eta_t$ (Definition 2.3). We will demonstrate that $E_t[A_t] \leq A_{t-1} + B_{t-1} - C_{t-1}$, for all $t$, in the Supplementary Material from which it follows that $A_t$ converges to zero with probability one and $\sum_{t=1}^\infty C_t < \infty$. We write

$$
\begin{aligned}
E_t\left[\|\boldsymbol{\theta}_t - \boldsymbol{\theta}^\star\|_2^2\right] &= E_t\left[\|\boldsymbol{\theta}_{t-1} - \eta_{t-1}\mathbf{Z}_{t-1} - \boldsymbol{\theta}^\star\|_2^2\right] \\
&= \|\boldsymbol{\theta}_{t-1} - \boldsymbol{\theta}^\star\|_2^2 - 2\eta_{t-1}E_t[(\boldsymbol{\theta}_{t-1} - \boldsymbol{\theta}^\star)^T\mathbf{Z}_{t-1}] + \eta_{t-1}^2 E_t[\|\mathbf{Z}_{t-1}\|_2^2] \\
&\leq \|\boldsymbol{\theta}_{t-1} - \boldsymbol{\theta}^\star\|_2^2 - 2\eta_{t-1}(\boldsymbol{\theta}_{t-1} - \boldsymbol{\theta}^\star)^T\boldsymbol{G}_{t-1} + \delta^2\eta_{t-1}^2 + \delta^2\eta_{t-1}^2\|\boldsymbol{\theta}_{t-1} - \boldsymbol{\theta}^\star\|_2^2 \\
&\leq \|\boldsymbol{\theta}_{t-1} - \boldsymbol{\theta}^\star\|_2^2 - 2M\eta_{t-1}\|\boldsymbol{\theta}_{t-1} - \boldsymbol{\theta}^\star\|_2^2 + \delta^2\eta_{t-1}^2 + \delta^2\eta_{t-1}^2\|\boldsymbol{\theta}_{t-1} - \boldsymbol{\theta}^\star\|_2^2,
\end{aligned}
$$

and the result follows. $\square$

# B EXPERIMENTS: FURTHER RESULTS AND IMPLEMENTATION DETAILS

## B.1 RESULTS

The accuracy scores for all models (including the LSTM) on the sequence arithmetic tasks are shown in Table 3. This table repeats results shown in Table 1, except here we show additional rows representing models that use LSTM as $\vec{f}$. We chose accuracy (0-1 loss) to be consistent with Zaheer et al. (2017); here we report mean absolute error to evaluate the differences it makes on our results. These can be found in Tables 4 and 5. The message is similar to the one told by accuracy scores; there is a drop in the mean absolute error as the value of $k$ increases and when using more sampled permutations at test-time (e.g., Janossy-20inf-LSTM versus Janossy-1inf-LSTM). Again, the power of using an RNN for $\vec{f}$ and training with $\pi$-SGD is salient on the variance task where it is important to exploit dependencies in the sequence. Beyond the performance gains, we also observe a drop in variance when sampling more permutations at test time. Furthermore, as discussed in the implementation section, we constructed $k$-ary models to have the same number of parameters regardless of $k$ for the results reported in the main body. We show results where this constraint is relaxed in Table 6. Here we see a modest improvement of $k$-ary models which stands to reason considering the embedding dimension fed to the Janossy pooling layer was reduced from 100 with $k = 1$ to 33 with $k = 3$ (please see the implementation section for details).

For the graph tasks, the plot of performance as a function of number of inference-time permutations is shown in Figure 2.

Table 3: Full table showing the Accuracy (and RMSE for the *variance* task) for all models used for the sequence arithmetic tasks. The *method* column refers to the method used to deal with the sum over all permutations. *Infr sample* refers to the number of permutations sampled at test time to estimate equation 11 for methods learned with $\pi$-SGD. $k = 1$ corresponds to DeepSets. tanh activations are used with the MLP. Standard deviations computed over 15 runs are shown in parentheses.

| $\vec{f}$ | method | infr sample | $k$ | $\rho$ | sum | range | unique sum | uniq. count | variance |
|---|---|---|---|---|---|---|---|---|---|
| MLP (30) | exact | – | 1 | Linear | 1.00(0.00) | 0.04(0.00) | 0.07(0.00) | 0.36(0.01) | 119.05(1.29) |
| MLP (30) | exact | – | 2 | Linear | 0.99(0.00) | 0.09(0.00) | 0.17(0.00) | 0.74(0.03) | 4.37(0.50) |
| MLP (30) | exact | – | 3 | Linear | 0.99(0.00) | 0.21(0.00) | 0.44(0.02) | 0.89(0.04) | 8.99(0.99) |
| MLP (30) | exact | – | 1 | MLP (100) | 1.00(0.00) | 0.97(0.01) | 1.00(0.00) | 1.00(0.00) | 1.95(0.24) |
| MLP (30) | exact | – | 2 | MLP (100) | 1.00(0.00) | 0.97(0.01) | 1.00(0.00) | 1.00(0.00) | 3.49(0.48) |
| MLP (30) | exact | – | 3 | MLP (100) | 0.93(0.02) | 0.93(0.02) | 1.00(0.00) | 1.00(0.00) | 6.90(0.47) |
| LSTM(50) | $\pi$-SGD | 1 | $|h|$ | Linear | 0.99(0.00) | 0.95(0.01) | 1.00(0.00) | 1.00(0.00) | 1.65(0.22) |
| LSTM(50) | $\pi$-SGD | 20 | $|h|$ | Linear | 0.99(0.00) | 0.97(0.01) | 1.00(0.00) | 1.00(0.00) | 1.39(0.26) |
| GRU(80) | $\pi$-SGD | 1 | $|h|$ | Linear | 0.99(0.01) | 0.98(0.00) | 1.00(0.00) | 1.00(0.00) | 1.43(0.23) |
| GRU(80) | $\pi$-SGD | 20 | $|h|$ | Linear | 0.99(0.00) | 0.99(0.00) | 1.00(0.00) | 1.00(0.00) | 1.20(0.23) |
| LSTM(50) | $\pi$-SGD | 1 | $|h|$ | MLP (100) | 0.99(0.01) | 0.99(0.00) | 1.00(0.00) | 1.00(0.00) | 1.05(0.77) |
| LSTM(50) | $\pi$-SGD | 20 | $|h|$ | MLP (100) | 0.99(0.00) | 1.00(0.00) | 1.00(0.00) | 1.00(0.00) | 1.02(0.41) |
| GRU(80) | $\pi$-SGD | 1 | $|h|$ | MLP (100) | 0.99(0.00) | 1.00(0.00) | 1.00(0.00) | 1.00(0.00) | 0.42(0.62) |
| GRU(80) | $\pi$-SGD | 20 | $|h|$ | MLP (100) | 0.99(0.00) | 1.00(0.00) | 1.00(0.00) | 1.00(0.00) | 0.40(0.37) |

Table 4: Mean Absolute Error of various Janossy pooling approximations under distinct tasks. The column method refers to the tractability strategy. Inf sample refers to the number of permutations sampled to estimate equation 11 for methods learned with $\pi$-SGD. $k = 1$ corresponds to DeepSets. tanh activations are used with the MLP's. Standard deviations computed over 15 runs are shown in parentheses.

| $\vec{f}$ | method | inf sample | $k$ | $\rho$ | sum | range | unique sum | unique count |
|---|---|---|---|---|---|---|---|---|
| MLP (30) | exact | – | 1 | Linear | 0.000(0.000) | 9.366(0.094) | 4.209(0.025) | 0.828(0.008) |
| MLP (30) | exact | – | 2 | Linear | 0.006(0.011) | 4.143(0.041) | 1.968(0.016) | 0.277(0.029) |
| MLP (30) | exact | – | 3 | Linear | 0.037(0.031) | 2.307(0.074) | 0.730(0.040) | 0.114(0.040) |
| MLP (30) | exact | – | 1 | MLP (100) | 0.001(0.000) | 0.033(0.003) | 0.000(0.000) | 0.000(0.000) |
| MLP (30) | exact | – | 2 | MLP (100) | 0.007(0.005) | 0.038(0.006) | 0.000(0.000) | 0.000(0.000) |
| MLP (30) | exact | – | 3 | MLP (100) | 0.091(0.026) | 0.147(0.049) | 0.000(0.000) | 0.000(0.000) |
| LSTM(50) | $\pi$-SGD | 1 | $|h|$ | Linear | 0.003(0.002) | 0.051(0.010) | 0.000(0.000) | 0.000(0.000) |
| LSTM(50) | $\pi$-SGD | 20 | $|h|$ | Linear | 0.001(0.001) | 0.035(0.006) | 0.000(0.000) | 0.000(0.000) |
| GRU(80) | $\pi$-SGD | 1 | $|h|$ | Linear | 0.007(0.012) | 0.020(0.005) | 0.000(0.000) | 0.000(0.000) |
| GRU(80) | $\pi$-SGD | 20 | $|h|$ | Linear | 0.001(0.002) | 0.014(0.004) | 0.000(0.000) | 0.000(0.000) |
| LSTM(50) | $\pi$-SGD | 1 | $|h|$ | MLP (100) | 0.007(0.010) | 0.006(0.001) | 0.000(0.000) | 0.000(0.000) |
| LSTM(50) | $\pi$-SGD | 20 | $|h|$ | MLP (100) | 0.004(0.006) | 0.005(0.001) | 0.000(0.000) | 0.000(0.000) |
| GRU(80) | $\pi$-SGD | 1 | $|h|$ | MLP (100) | 0.002(0.004) | 0.002(0.001) | 0.000(0.000) | 0.000(0.000) |
| GRU(80) | $\pi$-SGD | 20 | $|h|$ | MLP (100) | 0.002(0.003) | 0.002(0.001) | 0.000(0.000) | 0.000(0.000) |

Table 5: Mean Absolute Error of various Janossy pooling approximations for the variance task. The column method refers to the tractability strategy. Inf sample refers to the number of permutations sampled to estimate equation 11 for methods learned with $\pi$-SGD. $k = 1$ corresponds to DeepSets. tanh activations are used with the MLP's. Standard deviations computed over 15 runs are shown in parentheses.

| $\vec{f}$ | method | inf sample | k | $\rho$ | variance |
|---|---|---|---|---|---|
| MLP (30) | exact | – | 1 | Linear | 69.953(0.492) |
| MLP (30) | exact | – | 2 | Linear | 2.262(0.363) |
| MLP (30) | exact | – | 3 | Linear | 6.747(0.871) |
| MLP (30) | exact | – | 1 | MLP (100) | 0.613(0.107) |
| MLP (30) | exact | – | 2 | MLP (100) | 1.733(0.146) |
| MLP (30) | exact | – | 3 | MLP (100) | 4.379(0.318) |
| LSTM(50) | $\pi$-SGD | 1 | $\lvert h \rvert$ | Linear | 0.801(0.200) |
| LSTM(50) | $\pi$-SGD | 20 | $\lvert h \rvert$ | Linear | 0.698(0.412) |
| GRU(80) | $\pi$-SGD | 1 | $\lvert h \rvert$ | Linear | 0.795(0.205) |
| GRU(80) | $\pi$-SGD | 20 | $\lvert h \rvert$ | Linear | 0.672(0.332) |
| LSTM(50) | $\pi$-SGD | 1 | $\lvert h \rvert$ | MLP (100) | 0.604(0.078) |
| LSTM(50) | $\pi$-SGD | 20 | $\lvert h \rvert$ | MLP (100) | 0.422(0.102) |
| GRU(80) | $\pi$-SGD | 1 | $\lvert h \rvert$ | MLP (100) | 0.594(0.634) |
| GRU(80) | $\pi$-SGD | 20 | $\lvert h \rvert$ | MLP (100) | 0.517(0.084) |

Table 6: Accuracy (and RMSE for the *variance* task) of $k-ary$ Janossy pooling approximations with the same input dimension as $k = 1$ under distinct tasks. The *method* column refers to the method used to deal with the sum over all permutations. *Infr sample* refers to the number of permutations sampled at test time to estimate equation 11 for methods learned with $\pi$-SGD. $k = 1$ corresponds to DeepSets. tanh activations are used with the MLP. Standard deviations computed over 15 runs are shown in parentheses.

| $\vec{f}$ | method | k | $\rho$ | sum | range | unique sum | uniq. count | variance |
|---|---|---|---|---|---|---|---|---|
| MLP (30) | exact | 1 | Linear | 1.00(0.00) | 0.04(0.00) | 0.07(0.00) | 0.36(0.01) | 119.05(1.29) |
| MLP (30) | exact | 2* | Linear | 1.00(0.00) | 0.09(0.00) | 0.18(0.00) | 0.74(0.03) | 0.71(0.04) |
| MLP (30) | exact | 3* | Linear | 1.00(0.00) | 0.22(0.00) | 0.51(0.01) | 0.98(0.00) | 1.54(0.99) |
| MLP (30) | exact | 1 | MLP (100) | 1.00(0.00) | 0.97(0.01) | 1.00(0.00) | 1.00(0.00) | 1.95(0.24) |
| MLP (30) | exact | 2* | MLP (100) | 1.00(0.00) | 0.99(0.00) | 1.00(0.00) | 1.00(0.00) | 2.65(0.50) |
| MLP (30) | exact | 3* | MLP (100) | 1.00(0.00) | 0.99(0.00) | 1.00(0.00) | 1.00(0.00) | 3.44(0.51) |

Table 7: Mean Absolute Error of $k - ary$ Janossy pooling approximations with the same input dimension as $k = 1$ under distinct tasks. The *method* column refers to the method used to deal with the sum over all permutations. *Infr sample* refers to the number of permutations sampled at test time to estimate equation 11 for methods learned with $\pi$-SGD. tanh activations are used with the MLP. Standard deviations computed over 15 runs are shown in parentheses.

| $\vec{f}$ | method | k | $\rho$ | sum | range | unique sum | uniq. count | variance |
|---|---|---|---|---|---|---|---|---|
| MLP (30) | exact | 1 | Linear | 0.00(0.00) | 9.37(0.09) | 4.21(0.03) | 0.83(0.01) | 69.95(0.49) |
| MLP (30) | exact | 2* | Linear | 0.00(0.00) | 4.12(0.05) | 1.95(0.01) | 0.29(0.03) | 0.46(0.04) |
| MLP (30) | exact | 3* | Linear | 0.00(0.00) | 2.31(0.04) | 0.64(0.02) | 0.02(0.00) | 1.09(0.11) |
| MLP (30) | exact | 1 | Linear | 0.00(0.00) | 0.03(0.00) | 0.00(0.00) | 0.00(0.00) | 0.61(0.10) |
| MLP (30) | exact | 2* | MLP (100) | 0.00(0.00) | 0.01(0.00) | 0.00(0.00) | 0.00(0.00) | 0.96(0.09) |
| MLP (30) | exact | 3* | MLP (100) | 0.02(0.00) | 0.02(0.00) | 0.00(0.00) | 0.00(0.00) | 1.39(0.12) |

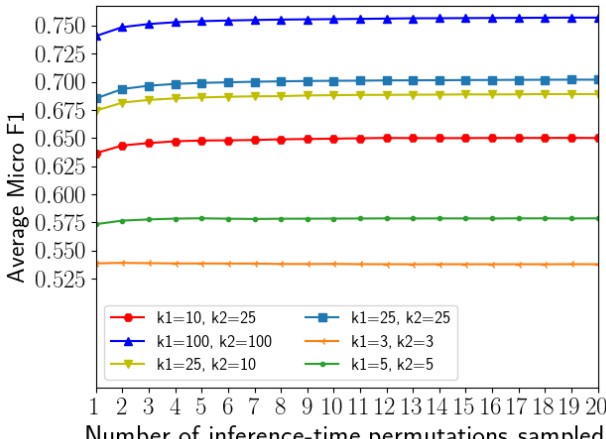

Figure 2: Mean performance vs number of permutations sampled at test time, PPI task

### B.2 IMPLEMENTATION AND EXPERIMENT DETAILS

**Sequence tasks** We extended the code from Zaheer et al. (2017), which was written in Keras(Chollet et al., 2015), and subsequently ported to PyTorch. For $k$-ary models with $k \in \{2, 3\}$, we always sort the sequence $\boldsymbol{x}$ beforehand to reduce the number of combinations we need to sum over. In the notation of Figure 1, $\boldsymbol{h}$ is an Embedding with dimension of floor$(\frac{100}{k})$ (to keep the total number of parameters consistent for each $k$ as discussed below), $\vec{f}$ is either an MLP with a single hidden layer or an RNN depending on the model ($k$-ary Janossy or full-Janossy, respectively), and $\rho$ is either a linear dense layer or one hidden layer followed by a linear dense layer. The MLPs in $\vec{f}$ have 30 neurons whereas the MLPs in $\rho$ have 100 neurons, the LSTMs have 50 neurons, and the GRUs have 80 hidden neurons. All activations are tanh except for the output layer which is linear. We chose 100 for the embedding dimension to be consistent with Zaheer et al. (2017).

For the $k$-ary results shown in the body, we made sure the number of parameters was consistent for $k \in \{1, 2, 3\}$ (see Table 8). We unify the number of parameters by adjusting the output dimension of the embedding. We also experimented with relaxing the restriction that $k$-ary models have the same numbers of parameters (Table 6), and the numbers of parameters in these models is also shown in Table 8. For the LSTM than GRU models, we follow the choice of Zaheer et al. (2017) which also reports that the choices were made to keep the numbers of parameters consistent.

Optimization is done with Adam (Kingma & Ba, 2015) with a tuned the learning rate, searching over $\{0.01, 0.001, 0.0001, 0.00001\}$. Training was performed on GeForce GTX 1080 Ti GPUs.

**Graph-based tasks** The datasets used for this task are summarized in Table 9. Our implementation is in PyTorch using Python 2.7, following the PyTorch code associated with Hamilton et al. (2017). That repo did not include an LSTM aggregator, so we implemented our own following the TensorFlow implementation of GraphSAGE, and describe it here. At the beginning of every forward pass, each vertex $v$ is associated with a $p$-dimensional vertex attribute $\boldsymbol{h}$ (see Table9). For every vertex in a batch, $k_1$ neighbors of $v$ are sampled, their order is shuffled, and their features are fed through an LSTM. From the LSTM, we take the short-term hidden state associated with the last element in the input sequence (denoted $\boldsymbol{h}_{(T)}$ in the LSTM literature, but this $\boldsymbol{h}$ is not to be confused with a vertex attribute). This short-term hidden state is passed through a fully connected layer to yield a vector of dimension $\frac{q}{2}$, where $q$ is a user-specified positive even integer referred to as the *embedding dimension*. The vertex's own attribute $\boldsymbol{h}$ is also fed forward through a fully connected layer with $\frac{q}{2}$ output neurons. At this point, for each vertex, we have two representation vectors of size $\frac{q}{2}$ representing the vertex $v$ and its neighbor multiset, which we concatenate to form an embedding of size $q$. This describes one convolution layer, and it is repeated a second time with a distinct set of learnable weights for the fully connected and LSTM layers, sampling $k_2$ vertices from each neighborhood and using the embeddings of the first layer as features. After each convolution, we may optionally apply a ReLU activation and/or embedding normalization, and we follow the decisions shown in the GraphSAGE code Hamilton et al. (2017). After both convolution operations, we apply a final fully connected layer to obtain the score, followed by a softmax (Cora, Pubmed) or sigmoid (PPI). The loss function is cross entropy for Cora and Pubmed, and binary cross entropy for PPI.

Table 8: Number of trainable parameters in each of the $k$-ary Janossy Pooling approaches. The $k$-ary models indicated with a $*$ take 100 dimensional embeddings as input to $\vec{f}$, in contrast with the approach taken in Table 1 where the embedding was of size floor$(100/k)$.

| $\vec{f}$ | $k$ | $\rho$ | # trainable parameters |
|---|---|---|---|
| MLP (30) | 1 | Linear | 3061 |
| MLP (30) | 2 | Linear | 3061 |
| MLP (30) | 3 | Linear | 3031 |
| MLP (30) | 2* | Linear | 6061 |
| MLP (30) | 3* | Linear | 9061 |
| MLP (30) | 1 | MLP (100) | 6231 |
| MLP (30) | 2 | MLP (100) | 6231 |
| MLP (30) | 3 | MLP (100) | 6201 |
| MLP (30) | 2* | MLP (100) | 9231 |
| MLP (30) | 3* | MLP (100) | 12231 |
| LSTM(50) | n | Linear | 30451 |
| GRU(80) | n | Linear | 43761 |
| LSTM(50) | n | MLP (100) | 35601 |
| GRU(80) | n | MLP (100) | 51881 |

Table 9: Summary of the graph datasets

| CHARACTERISTIC | CORA | PUBMED | PPI |
|---|---|---|---|
| Number of Vertices | 2708 | 19717 | 56944, 2373[a] |
| Average Degree | 3.898 | 4.496 | 28.8[a] |
| Number of Vertex Features | 1433 | 500 | 50 |
| Number of Classes | 7 | 3 | 121[b] |
| Number of Training Vertices | 1208 | 18217 | 44906[c] |
| Number of Test Vertices | 1000 | 1000 | 5524[c] |

[a] The PPI dataset comprises several graphs, so the quantities marked with an "a", represent the characteristic of the average graph .

[b] For PPI, there are 121 targets, each taking values in $\{0, 1\}$.

[c] All of the training nodes come from 20 graphs while the testing nodes come from two graphs not utilized during training.

The number of trainable parameters in each model is independent of $k_1$ and $k_2$ by the design of LSTMs (the same is true for the mean-pooling aggregator). The only variation in the number of weights is in the dimensions of the input and output features, which differ by dataset. Please see Table 10 for details.

Optimization is performed with the Adam optimizer (Kingma & Ba, 2015). The training routine for the smaller graphs (Cora, Pubmed) is not guaranteed to see the entire training data, in contrast with the scheme applied to the larger PPI graph. For Cora and Pubmed, we form 100 minibatches by randomly sampling subsets of 256 vertices from the training dataset (with replacement). With PPI, we perform 10 full epochs: at each epoch, the training data is shuffled, partitioned into minibatches of size 512, and we pass over each. In either case, the weights are updated after computing the gradient of the loss on each minibatch.

The hyperparameters were set by following Hamilton et al. (2017); no hyperparameter optimization was performed. For every dataset, the embedding dimension was set to $q = 256$ at both conv layers. For Pubmed and PPI, the learning rate is set at 0.01 while for Cora it is set at 0.005.

At test time, we load the weights obtained from training, perform 20 forward passes – which shuffles the input sequence by design – average the predicted probabilities (i.e. softmax output) from each forward pass, and choose the class that maximizes the averaged probabilities.

The implementation for Mean Pooling is similar in spirit but replaces $\vec{f}$ with a permutation invariant function. The details can be found by viewing our repo on GitHub.

Table 10: Number of trainable parameters for each model in the graph task. The number does not depend on $k_1$ or $k_2$.

| $\vec{f}$ | CORA | PUBMED | PPI |
|---|---|---|---|
| mean-pool | 400512 | 161152 | 61056 |
| LSTM | 2541440 | 1465600 | 977408 |

## C  LATEX FOR JANOSSY FUNCTION MARKERS

The commands below can be directly copied and pasted into LATEXsource to create the Janossy function markers. Please use the `amsmath` package.

To typeset $\overline{\overline{f}}$, we define in LATEXas

```
\newcommand*\dbar[1]{\overline{\overline{\lower0.2ex\hbox{$#1$}}}}
```

and type

```
$\dbar{f}$.
```

Similarly, for $\vec{f}$, we define in LATEXas

```
\newcommand{\harrow}[1]{\mathstrut\mkern2.5mu#1\mkern-11mu\raise1.6ex
\hbox{$\scriptscriptstyle\rightharpoonup$}}
```

and type

```
$\harrow{f}$.
```

Last, the definition above for $\vec{f}$ caused difficulties in environments such as `figure`, so we defined and occasionally used in LATEX

```
\newcommand{\harrowStable}[1]{\overset{\rightharpoonup}{#1}}.
```

