# OpenReview forum: "Janossy Pooling: Learning Deep Permutation-Invariant Functions for Variable-Size Inputs"
_ICLR.cc/2019/Conference_

### Official Review · AnonReviewer1 · 2018-11-02
**Interesting demonstration that standard pooling methods are insufficiently flexible**

**Rating:** 8
**Confidence:** 4

**Review:**

I really enjoyed this paper. It takes an idea which at first glance seems to be obviously bad (if you want permutation invariance, build a model that considers all permutations) and uses it to make the important point that the universal approximation results contained in Deep Sets [Zaheer et al. 2017] are not the last word on pooling. Janossy Pooling is intractable for most problems of interest (because it sums over all n! permutations of the input set) so the authors suggest 3 tractable alternatives: canonical orderings, k-ary dependencies and SGD / sampling-based approaches. Only the latter two are explored in detail, so I’ll focus on them:

K-ary dependencies
Functions that are restricted to k-ary dependencies in Janossy Pooling require summing over only |h|! / (|h| - k)! terms - that is they sum over the permutations of subsets of h of length k. In the experimental section, the authors show that this recovers some of the performances lost by using sum / mean pooling (as in Deep Sets), but this suggests the natural question: is it the fact that you’re explicitly modelling higher-order interactions that improves performance? Or is it that you’re doing Janossy pooling over the higher order interactions (i.e. summing over permutations of non-invariant functions)?

These two effects could be separated by comparing to invariant models that allow higher order interactions. E.g. you could compare against Santoro et al. [2017] who explicitly model pairwise interactions (or similarly any of the graph convolutional models [Kipf and Welling 2016, Hamilton et al 2017, etc.] with a fully connected graph would do the same); similarly Hartford, et al. [2018] allow for k-wise interactions by extending Deep Sets to exchangeable tensors - the permutation invariant analog of k-ary Janossy Pooling. All of these approaches model k-wise interactions through sum-pooling over permutation invariant functions so this lets you address the question - is it the permutation invariance that’s the problem (necessitating k-ary Janossy pooling) or is it the lack of higher-order interaction terms?

SGD approaches:
I think that the point that the sampling-based approaches are bias with respect to the Janossy sum is important to make and I liked the discussion around it, but I don’t follow the relevance of Proposition 2.2? I see that it gives conditions under which we can expect \pi-SGD to converge, but we aren’t provided with any guidance about how likely those conditions are to be satisfied? Furthermore - these conditions don’t seem to be specific to \pi-SGD - any SGD algorithm with “slightly biased” gradients that satisfy these conditions would converge. The regularization idea is interesting, but it isn’t evaluated so we’re left with theory that doesn’t provide guidance and isn’t evaluated.

Summary:
There are two ways to read this paper:
 1. Janossy pooling as a framework & proposed pooling approach implemented in one of the two ways discussed above.
 2. Janossy pooling as an intractable upper bound on what we might want from a pooling method (with approximations in the form of the LSTM approaches) and a demonstration that our current invariant pooling methods are insufficient.

I liked the paper based on reading (2). Janossy pooling clearly demonstrates limitations to sum / mean pooling which is widely used in practice which shows the need for better alternatives and it is on this basis that I’m arguing for it’s acceptance. My view is that the experimental section is too limited to support reading (1) which asserts that k-ary pooling or LSTM + sampling approaches are the right solution to this problem.

[Zaheer et al. 2017] - Manzil Zaheer, Satwik Kottur, Siamak Ravanbakhsh, Barnabas Poczos, Ruslan Salakhutdinov, and
Alexander Smola. Deep Sets
[Santoro et al. 2017] - Adam Santoro, David Raposo, David G Barrett, Mateusz Malinowski, Razvan Pascanu, Peter Battaglia, and Tim Lillicrap. A simple neural network module for relational reasoning.
[Kipf and Welling 2016] - Thomas N. Kipf and Max Welling. Semi-Supervised Classification with Graph Convolutional Net- works
[Hamilton et al 2017] - William L. Hamilton, Rex Ying, and Jure Leskovec. Inductive Representation Learning on Large Graphs
[Hartford, et al. 2018] - Jason S. Hartford, Devon R. Graham, Kevin Leyton-Brown, and Siamak Ravanbakhsh. Deep models of interactions across sets

---

> ### Comment · AnonReviewer1 · 2018-11-14
> **Unique sum and unique count require a multiset not a set**
>
> I didn't pick this up in my initial review, but the "unique sum" and "unique count" tasks in the synthetic experiments go beyond the scope of the deep sets work since that paper refers to sets not multisets. "Unique" doesn't make sense for sets. These experiments should be removed (or at the very least this should be made clear). Similarly for the other tasks, sampling without replacement makes more sense to ensure the input is in fact a set.

---

> > ### Author Response · Authors · 2018-11-16
> > **Clarification regarding 'unique count' and 'unique sum' tasks**
> >
> > Our primary interest was in permutation-invariant functions, and we only used the term “set function” to follow Zaheer 2017.  Please note that Deep Sets performs the sum task on “sets” of integers {0, 1, 2, …, 9} of size 50 which must have duplicates.  In following their design, our input sequences also have duplicates.
> >
> > We also note that extensions of the Deep Sets theorem relating permutation-invariance and sum pooling was recently extended to include multisets in [Xu et al 2018].
> >
> > We will add a line to the paper to clarify this.  Thanks.
> >
> > [Xu et al 2018] Xu, Keyulu, Weihua Hu, Jure Leskovec, and Stefanie Jegelka. "How Powerful are Graph Neural Networks?." arXiv preprint arXiv:1810.00826 (2018).

---

> ### Author Response · Authors · 2018-11-18
> **Response to Reviewer1**
>
> Thank you for your positive comments.  We also see Janossy Pooling as simultaneously providing theory that highlights limitations of existing methods and an overarching framework for developing pooling functions.  We also agree that more experiments (as always) are beneficial; our revision includes a more thorough experiments section upon which we elaborate below.
>
> (1) In the experimental section, the authors show that [k-ary Janossy Pooling] recovers some of the performances lost by using sum / mean pooling...Is it the fact that you're explicitly modelling higher-order interactions that improves performance? Or is it that you're doing Janossy pooling over the higher order interactions (i.e. summing over permutations of non-invariant functions)?
>
> Our development of k-ary Janossy Pooling (JP) demonstrates that the increased performance associated with k>1 does not rely upon using permutation-sensitive Janossy functions \harrow{f}. Indeed, our proof that (k-1)-ary JP is less expressive than k-ary JP constructs a permutation-invariant k-ary Janossy function (\harrow{f}) which cannot be expressed by any (k-1)-ary Janossy function, permutation-sensitive or otherwise.  We modeled \harrow{f} as permutation-sensitive in our experiments since basic neural network building blocks are permutation-sensitive.
>
> (2) I don't follow the relevance of Proposition 2.2? I see that it gives conditions under which we can expect \pi-SGD to converge, but we aren't provided with any guidance about how likely those conditions are to be satisfied? Furthermore - these conditions don't seem to be specific to \pi-SGD - any SGD algorithm with ``slightly biased'' gradients that satisfy these conditions would converge.
>
> We sought to reassure the reader of the appropriateness of randomly sampling and forwarding just one permutation of the sequence during training -- which at first glance may appear inappropriate.  We agree that this can be achieved simply by pointing out the similarity to ''typical'' SGD and we have revised our paper accordingly and moved the detailed proof to the appendix.
>
> (3) My view is that the experimental section is too limited to support reading (1) which asserts that k-ary pooling or LSTM + sampling approaches are the right solution to this problem.
>
> While we agree that reading (2) is our preferred reading too, we have added experiments to the revised version which provide further support of the power of proposed JP models.  These include (a) the addition of a more complex \rho (the function composed with the output of pooling) to all models for the arithmetic tasks, which presents a more competitive baseline, (b) the addition of a harder arithmetic task -- computing the variance of a sequence of integers -- and (c) further analysis of the impact of increasing the number of permutations sampled at test-time for prediction in a pi-SGD model.  The latter was performed on the PPI graph, a new dataset evaluated in this submission.
>
> (a and b) Whereas \rho was previously a linear layer only, we have added results where \rho is an MLP with a single hidden layer.  Our results show that Janossy pooling architectures achieve superior or similar performance to the baseline of sum pooling across all tasks -- including the variance task -- for either choice of \rho.  Notice that the GRU model with an MLP \rho achieves a mean RMSE of 0.40 on the variance task, beating the sum-pooling baseline by a substantial margin.
>
> (c) Our arithmetic tasks show a clear benefit of averaging over more permutations at test time; doing so either improved the mean performance or left it unchanged (especially when performance was already saturated).  We have also expanded our investigation of this phenomenon in the graphs tasks, where we plotted performance as a function of the number of permutations sampled  at test time across different models.  We saw that simply sampling just a few permutations led to consistent and statistically significant gains in performance.  These gains level off but do not degrade as more permutations are sampled.

---

### Official Review · AnonReviewer2 · 2018-11-05
**A few of comments and request for clarifications**

**Rating:** 5
**Confidence:** 4

**Review:**

In this paper, the authors presented a new pooling method called Janossy Pooling (JP), which is designed to better capture high-order information by addressing two limitations of existing works - fixed pooling function and fixed-size inputs. The studied problem is important and the motivation is clear, where the inputs are sets of objects such as values or vectors and how we can learn a good aggregation function to maximally preserve the information in the original sequence. The authors attacked this problem by firstly formally formulating this problem and introducing a general approach as well as a few of approximation methods to realize it in practice. They also discussed the connections of this work and some existing works such as deep set, which I found is quite useful.

In general, JP was proposed to learn permutation-invariant function for aggregating the information of the input sequence. The basic idea of JP is to simply take all generated order of sequences from the original sequence input, which however I found is not new since it has been conceptually discussed already in the literature.  Since this approach is computationally prohibitive, there are several ways of approximations to approach the solution. As the authors are aware of the existing works in the literature, these approaches were discussed before either in the same context or in some particular learning tasks. From this perspective, the proposed solutions are not novel either.

The experimental results are particularly weak. It is little interesting on the first toy problem and the results on graph embedding are not promising. In Table 2, it is clearly shown that the LSTM aggregation functions on the randomly sampled sequences are really beating the simple mean aggregation function. I think the authors need much more experiments to demonstrate why we need LSTM based pooling for realizing JP in terms of both the final accuracy and computational cost.

-------------------------------------------------
After reading the authors' rebuttals, they have addressed part of my concerns but I still think the current form is not below the acceptance threshold due to its weak experimental results and unclear technical details.

---

> ### Author Response · Authors · 2018-11-18
> **Response to Reviewer2**
>
> Thank you for your comments. We address the two issues, experimental evaluation and novelty, below.
>
> 1) "Experiments weak"
>
> (1.a) "Toy problems:" Our arithmetic tasks are more challenging than those of Deep Sets, whose experimental methodology we extend. That paper evaluated the model on a task that adds a set of digits. While summation does not require exploiting dependencies among elements in the sequence, we consider tasks such as “range” where doing so is imperative.  Our updated submission adds the task of computing the variance of a sequence of 10 integers. This update also makes our former preliminary results part of the main paper with a discussion of the new insights (here summarized in points 1.b and 1.c.)
>
> Overall, our work is focused on generalizing today's pooling methods rather than any specific task. With that view, we make our tasks as simple as they can be to avoid spurious effects, but not so simple that the effects of increased pooling expressiveness do not apply. We welcome suggestions of ways to improve them.
>
> (1.b) "No significant gains". "LSTM does not beat mean pooling."  The new variance task shows pi-SGD + GRUs + MLP \rho significantly outperforms other methods, including sum-pooling (variance requires better modeling of high-order interactions). Overall, pi-SGD + GRUs + MLP \rho  yields equal or superior performance across all arithmetic tasks. In general, using RNNs has the benefits of accepting variable-length sequences and seamlessly exploiting dependencies within the sequence.
>
> (1.c) "Graph tasks": We followed the tasks found in Hamilton et al. (2017), which we found are quite easy. Our main interest was in evaluating differences between the different JP approaches (different choices of k and the impact of proper inference) on a task distinct from the arithmetic ones, and the results confirmed the anticipated benefits of using better inference at test time. In particular, proper inference of the \pi-SGD + LSTM model via Remark 2.2 can yield performance gains "for free" simply by averaging over forwarded permutations of the input sequence at test time.
>
> 2) "Novelty": "The basic idea of JP is to simply take all generated order of sequences from the original sequence input, which however I found ... has been conceptually discussed already in the literature." We will try to answer this in a few different ways.
>
> (2.a) "There is prior modeling work summing permutations in pooling layers". To the best of our knowledge, our pooling framework is the first that generalizes pooling, unless the reviewer is aware of other work we haven't cited. As we answer next, Hamilton et al. (2017) and Moore & Neville (2017) performed pi-SGD in ad hoc manner, at the time it was not clear that it was a sound optimization procedure. Our work provides the theoretical underpinnings for their approach.
>
> (2.b) "\pi-SGD is not novel because it has already been tried". Hamilton et al. (2017) and Moore & Neville (2017) did not provide a theoretical justification for their approach, and it was not obvious how to extend it. Our framework provides a theoretical justification for why and how pi-SGD works (SGD on the aforementioned ideal), as well as a characterization of the *correct* way to do inference at test time (missing in Hamilton et al. (2017)).
>
> (2.c) "Novelty of k-ary Janossy pooling". To the best of our knowledge, k=3,... in full generality has not been tried. Deep Sets (with k=1) shows that sum-pooling is a universal approximator to permutation-invariant functions if the upper layers are universal approximators. We show that if the upper layers are not universal approximators (or if the universal approximation is hard to learn), k-ary Janossy pooling (k > 1) is more powerful. Moreover, we show that this pooling approach is equivalent to summing over permutation-sensitive functions and achieves tractability via a restricted model class (functions with k inputs) rather than an approximate algorithm. Thus, our framework links two views of pooling: Inductive biases imposed on the model to capture dependencies in the sequence is inextricably linked with tractability strategies and present a tradeoff with learnability.

---

### Official Review · AnonReviewer4 · 2018-11-08
**Emergency review for Janossy Pooling**

**Rating:** 7
**Confidence:** 4

**Review:**

I have found the ideas proposed in the paper very insightful and interesting. The paper, in general, is written very well and is accessible.  My most important concern is

 The whole development seems not as effective as k =1 in Table.2 (BTW, there is a typo there). One wonders, why for k =2, k =1 is not included? That is, can the formulation be changed in a way that \downarrow operator represents l \in {1 \cdots k} projections?  In the end, the method creates k tuples and feed them through specific fs so why not having smaller tuples?

The rest of my review below hopefully can help improving the paper;


- Is there any reason as to why higher order Janossy poolings do not perform as good as k =1 for the sum experiment?

- Can you report the number of parameters for the developments (Janossy -k)? Some examples according to the experiments help.

- I am a bit lost to grasp the paragraph below Eq.4, can you rephrase it and possibly provide references?

- When it comes to testing, how do you use Eq.13? Do you sample a few permutation and compute 13? If yes, how many in practice?

- In preposition 2.1,  n seems confusing, why not |h|

- In P6, x_i is a sequence. this needs to be mentioned

---

> ### Author Response · Authors · 2018-11-18
> **Response to Reviewer4**
>
> Thank you for your positive comments. We address your concerns below.
>
> "- Is there any reason as to why higher order Janossy poolings do not perform as good as k =1 for the sum experiment?"
>
> The sum task is an easy task, designed for k=1. Our revised manuscript shows sum task results with more runs and more epochs and the difference is not statistically significant.
>
> “- The whole development seems not as effective as k =1 in Table.2....”
>
> Theorem 2.1 shows that Janossy Pooling (JP) with k-ary dependencies includes and is more expressive than JP with (k-1)-ary dependencies, but there will be tasks where it is sufficient to let k=1 (and also easier to optimize).  This is especially true for easy tasks like the sum task which do not require exploiting dependencies within the input sequence.  Our revised manuscript now considers the harder task of computing the variance of a sequence of numbers. For this harder task, full-sequence Janossy (k = |h|) is significantly more accurate than k = 1,2,3, by using pi-SGD to train the model (which optimizes \doublebar{J} rather than \doublebar{L}). In the range task, full Janossy (k = |h|) + GRU + pi-SGD also shows significant gains over k=1,2,3. For all other tasks, Janossy k =|h| + GRU + pi-SGD performs as well as the other approaches.
>
> "- One wonders, why for k =2, k =1 is not included? That is, can the formulation be changed in a way that \downarrow operator represents l \in {1 \cdots k} projections?  In the end, the method creates k tuples and feed them through specific fs so why not having smaller tuples?"
>
> Theoretically it is not necessary (by Theorem 2.1) but is an interesting direction for future work that could help in practice. It is clear, however, that Janossy k = |h| with GRU + pi-SGD is hard to beat in more challenging tasks.
>
> "-Can you report the number of parameters for the developments (Janossy -k)? Some examples according to the experiments help. "
>
> We have added the number of parameters in the Supplementary Material (Table 7 and Table 9) together with more details about our experimental setting. We have also tested k=2,3 with more complex models for \arrow{f}, the Supplementary Material shows the improved results.
>
> "- I am a bit lost to grasp the paragraph below Eq.4, can you rephrase it and possibly provide references?"
>
> Thank you, we rephrased our observations to simplify the exposition. We also considered the pros and cons of including a proof that Eq.4 captures any permutation-invariant function with an expressive-enough set of permutation-sensitive functions: the proof is straightforward as one can simply add all possible asymmetries (that cancel out when summing over all permutations) to the set of all permutation-invariant functions and make this a set of permutation-sensitive functions. It could be useful as a Proposition but, given the page limit, we have chosen to omit this straightforward proof in favor of other observations.
>
> “- When it comes to testing, how do you use Eq.13? Do you sample a few permutation and compute 13? If yes, how many in practice?”
>
> We have rewritten our experimental section to clarify how Eq.13 is used. We recommend looking at the new Table 1 which now more clearly defines "infr samples" to describe how many samples we use to estimate Eq.13.
>
> " - In preposition 2.1, n seems confusing, why not |h| "
>
> That was a typo, we have changed to |h|. Thank you!
>
> - In P6, x_i is a sequence. this needs to be mentioned
>
> Thank you. We have made changes in the notation to clarify that x(i) is the i-th sequence from the training (test) data.

---

### Public Comment · (anonymous) · 2018-11-19
**Question about the second experiment.**

Thank you. I also really enjoyed reading the paper.
For the vertex classification, without the aggregating function, minibatch sampling approaches (e.g. FASTGCN) are also getting almost similar or better accuracy than exact mean-pool (GRAPHSAGE) according to the paper. FASTGCN samples a few nodes to learn the graph neural network, so it is already fast and can avoid the additional computational burden of the aggregator. This means that the need for aggregation function is not clear for the GCN-baed models. I think more analysis with variance or other better measures are needed to show why it is meaningful for the vertex classification task as well.

 It may be useful to test it with the work (Moore & Neville, 2017).

(Chen, Ma, and Xiao, 2018) Jie Chen, Tengfei Ma, Cao Xiao. FASTGCN: Fast Learning with graph convolutional networks via importance sampling.
(Moore & Neville, 2017) John Moore and Jennifer Neville. Deep collective inference.

---

> ### Author Response · Authors · 2018-11-21
> **Response to question about the second experiment.**
>
> Thank you, the comment raises an interesting point.
>
> 0. GraphSAGE is an instantiation of Janossy Pooling (JP) -- k-ary JP trained with pi-SGD (see our remark "Combining pi-SGD and Janossy with k-ary Dependencies" in our revised submission). This was the motive behind our second set of experiments, where we study the impact of proper test-time inference, in contrast to the ad hoc LSTM aggregator of Hamilton et al., 2017. As predicted by our theory, proper inference led to significant gains in performance.
>
> 1. FASTGCN similarly falls under the framework of k-ary JP with pi-SGD optimization, applying importance sampling for variance reduction (the FASTGCN paper may be somewhat confusing to readers because it uses Lebesgue integrals and dP() measures instead of sums, but its Algorithm 2 is just an importance sampling procedure). In our submission we mention one interesting variance-reduction technique used by Zolna et al., 2018, but we now realize that it would be valuable to add additional traditional techniques such as importance sampling (used in FASTGCN), Rao-Blackwellization, and control variates to our discussion. Variance reduction helps the pi-SGD objective in equation 10 (\doublebar{J}) become more like the "original" objective in equation 6 (\doublebar{L}) (see the paragraph below Proposition 2.2).
>
> 2. We are excited by the prospect of applying the lessons learned from Janossy Pooling to Graph Neural Networks (GNNs) in our future work. We believe that these insights will inspire new forms of aggregation functions in GNNs. A promising avenue is to combine JP with other variance-reduction techniques such as layer normalization and ensemble techniques like dropout, which have demonstrated strong performance on GNN tasks like PPI (Chen and Zhu, 2017).
>
> 3. Please note that Hamilton et al., 2017 did not use exact mean pooling in their experiments but sampled a subset of neighbors. Our experiments deviate from theirs on this point. Also note that for Cora and PubMed, sampling 25 neighbors (as in our updated draft) effectively samples all neighbors, but not for PPI.
>
>
> (Chen and Zhu, 2017) - J. Chen and J. Zhu. Stochastic training of graph convolutional networks. arXiv preprint arXiv:1710.10568, 2017.
>
> (Zolna et al., 2018) - Konrad Zolna, Devansh Arpit, Dendi Suhubdy, and Yoshua Bengio. Fraternal dropout. ICLR, 2018.

---

> > ### Author Response · Authors · 2018-11-26
> > **Revised Submission**
> >
> > Thanks again for your question.  We have updated the submission to include the comments we made and a few other minor improvements.

---

### Meta-Review · Area_Chair1 · 2018-12-15
**Interesting take on permutation invariances.**

**Confidence:** 4
**Recommendation:** Accept (Poster)

**Metareview:**

AR1 is concerned about whether higher-order interactions are modeled explicitly and if pi-SGD convergence conditions can be easily satisfied. AR2 is concerned that basic JP has been conceptually discussed in the literature and \pi-SGD is not novel because it was realized by Hamilton et al. (2017) and Moore & Neville (2017). However, the authors provide some theoretical analysis for this setting in contrast to prior works. AR1 is also concerned that the effect of higher-order information has not been 'disentangled' experimentally from order invariance. AR4 is concerned about  poor performance of higher order Janossy pooling compared to k =1 case and asks about the number of hyper-parameters. The authors showed a harder task of computing the variance of a sequence of numbers in response.

On balance, despite justified concerns of AR2 about novelty and AR1 about experimental verification, the work appears to tackle an interesting topic.  Reviewers find the problem interesting and see some hope in the proposed solutions. On balance, AC recommends this paper to be accepted at ICLR. The authors are asked to update manuscript to reflect honestly weaknesses as expressed by reviewers, e.g. issue with effects of 'higher-order information' and 'disentangled' from order invariance.

---

> ### Author Response · Authors · 2019-02-26
> **Response to Metareview**
>
> We would like to thank the reviewers.  We have updated our paper to clarify a few points and address the few remaining reviewer concerns not addressed during the rebuttal phase. In particular, this version adds further refinements to: the verifiability of pi-SGD conditions, capturing higher-order dependencies with k-ary Janossy, and the applicability of Janossy pooling to multisets.